# A Survey on Graph Construction for Geometric Deep Learning in Medicine: Methods and Recommendations

Tamara T. Mueller[1]      Sophie Starck[1]      Alina Dima[1]      Stephan Wunderlich[2,3]
Kyriaki-Margarita Bintsi[4]      Kamilia Zaripova[5]      Rickmer Braren[2]
Daniel Rueckert[1,4]      Anees Kazi[6]      Georgios Kaissis[1,7]

[1] *AI in Medicine and Healthcare, Technical University of Munich, Germany*
[2] *Department for Interventional Radiology, Technical University of Munich*
[3] *Department for Radiology, Ludwig-Maximilians-University Munich*
[4] *BioMedIA, Imperial College London, UK*
[5] *Department for Computer Aided Medical Procedures and Augmented Reality, Technical University of Munich*
[6] *Laboratories for Computational Neuroimaging, Harward Medical School*
[7] *Machine Learning in Biomedical Imaging, Helmholtz Munich*
*Contact: {tamara.mueller; g.kaissis}@tum.de*

**Reviewed on OpenReview:** *https://openreview.net/forum?id=sWlHhfijcS*

## Abstract

Graph neural networks are powerful tools that enable deep learning on non-Euclidean data structures like graphs, point clouds, and meshes. They leverage the connectivity of data points and can even benefit learning tasks on data, which is not naturally graph-structured –like point clouds. In these cases, the graph structure needs to be determined from the dataset, which adds a significant challenge to the learning process. This opens up a multitude of design choices for creating suitable graph structures, which have a substantial impact on the success of the graph learning task. However, so far no concrete guidance for choosing the most appropriate graph construction is available, not only due to the large variety of methods out there but also because of its strong connection to the dataset at hand. In medicine, for example, a large variety of different data types complicates the selection of graph construction methods even more. We therefore summarise the current state-of-the-art graph construction methods, especially for medical data. In this work, we introduce a categorisation scheme for graph types and graph construction methods. We identify two main strands of graph construction: static and adaptive methods, discuss their advantages and disadvantages, and formulate recommendations for choosing a suitable graph construction method. We furthermore discuss how a created graph structure can be assessed and to what degree it supports graph learning. We hope to support medical research with graph deep learning with this work by elucidating the wide variety of graph construction methods.[8]

## 1 Introduction

Graphs can be used to represent several kinds of real-world datasets, such as networks, interactions, connections, or information flows. They hold information encoded in a set of nodes and edges, which connect pairs of nodes. They can add a structural component to otherwise independent data points. A wide variety of data can be structured as graphs, such as knowledge (Ruan et al., 2021), (3D) structures in space (Wolterink & Suk, 2021), brain signals (Kim et al., 2021), or maps (Yu et al., 2021c). Yet, the question of how to construct an appropriate graph structure from a given dataset can be non-trivial.

---

[8]The authors used ChatGPT for minor writing support.

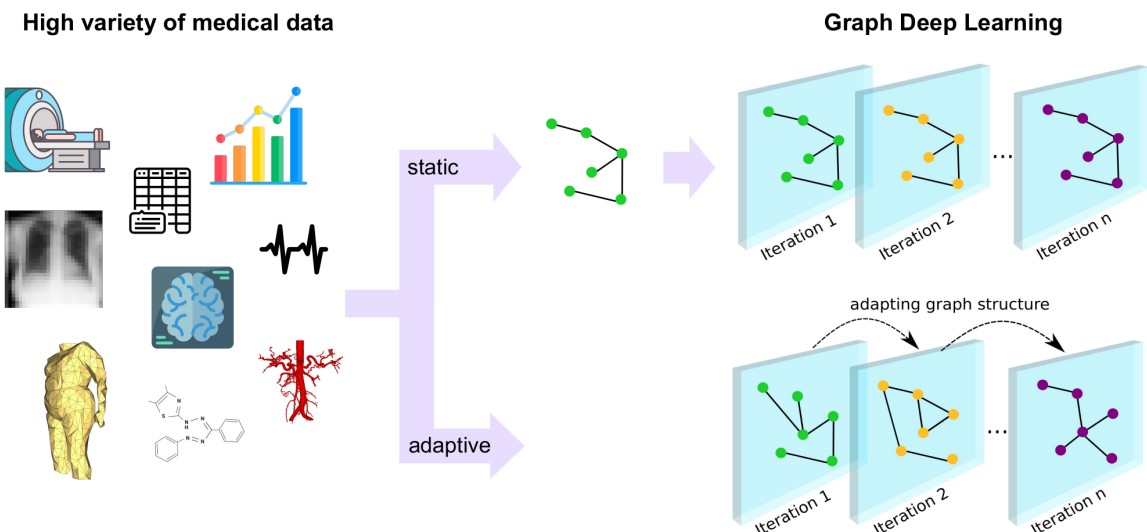

Figure 1: In this work, we summarise the state-of-the-art graph creation methods that allow one to transform a high variety of medical datasets into graph structures to perform graph deep learning. Static graph creation methods (upper row) extract the graph structure prior to learning, while adaptive methods (lower row) change the graph structure during training.

Graph learning techniques have been designed to apply deep learning (DL) methods directly to non-Euclidean datasets like graphs or meshes (Bronstein et al., 2017). These methods have since been frequently applied to data that can efficiently be structured by using a graph, for example, social networks (Fan et al., 2019) or molecules (Moreira-Filho et al., 2022), and have addressed tasks like friendship recommendations or drug discovery. Graph deep learning also naturally benefits many applications on medical datasets, as Graph Neural Networks (GNNs) have proven to be powerful algorithms for downstream tasks like medical diagnosis (Parisot et al., 2017), or image segmentation (Xie et al., 2022). They allow for straightforward integration of multi-modal data, such as image features and clinical data, into one coherent data structure, which has been explored in the context of so-called population graphs, where a medical cohort is represented by a graph structure instead of a tabular database (Kazi et al., 2022; Parisot et al., 2017).

Even in cases where a graph representation is not the default choice for a dataset, it has been shown that imposing such a graph structure by leveraging connections between data points can improve the performance of ML algorithms (Parisot et al., 2017; Ahmedt-Aristizabal et al., 2021; Bessadok et al., 2022; Pellegrini et al., 2022). This way, relations can be utilised or newly discovered, which can be beneficial for the task at hand (Cosmo et al., 2020). The application of GNNs to point cloud datasets, for example, can improve model performance compared to only using individual data points (Wang et al., 2019). This has also been shown in the medical field for tasks like disease prediction (Parisot et al., 2017), vessel segmentation (Paetzold et al., 2021), or the interaction between symptoms, diseases, and medication (Ruan et al., 2021). Here, spatial proximity (Yao et al., 2022; Hansen & Heinrich, 2021), medical knowledge (Mueller et al., 2022a), anatomical structures (Sun et al., 2021), correlations (Kim et al., 2021), or cartographic location (Yu et al., 2021c), have been used to generate a graph structure from previously disconnected data points.

This additional processing step of generating a graph structure introduces new challenges to the overall downstream task (Ahmedt-Aristizabal et al., 2021) and the definition or fine-tuning of nodes and edges hold a range of crucial design choices. This has turned out to be especially challenging in many medical settings since for medical images or health reports a graph structure is not the default choice of representation. Brain connectivity graphs have, for example, shown to be a suitable representation of the human connectome (Bessadok et al., 2022), which represents a map of neural connections in the brain. However, the construction of the brain graph holds challenges like the temporal component of functional magnetic resonance imaging (fMRI). Tube-like structures like airways (Selvan et al., 2020) and vessels (Paetzold et al., 2021) can be accurately represented by a graph that follows the anatomy of the structure at hand. Still, the concrete

extraction of the graph structure requires precisely segmenting the curvilinear structures or transforming branching points into nodes.

In addition to the wide variety of graph construction methods, the strong impact of the graph structure on the success of the learning task makes this especially challenging (Luan et al., 2022). We thus conclude that the construction of a suitable graph structure is crucial to optimally leverage the connectedness inherent to the dataset.

## 1.1 Why the graph construction matters

Since the introduction of GNNs, many works have shown that GNNs can improve performance on non-Euclidean datasets compared to graph-agnostic DL models (Cosmo et al., 2020; Parisot et al., 2017; Ahmedt-Aristizabal et al., 2021). However, this assumption does not apply to all settings and datasets and recent works have demonstrated that GNNs outperform graph-agnostic models only under specific circumstances. This can often be attributed to the utilisation of unsuitable graph structures and can even lead to simple graph-agnostic methods outperforming GNNs (Luan et al., 2022; Zhu et al., 2020). One of the reasons for this might be an over-smoothing of node features over unideal neighbourhoods (with, for example, highly diverse labels), which complicates the establishment of suitable feature embeddings required for the downstream task. When node features of neighbours with highly different labels (and therefore different node features) get averaged during message passing, the resulting node embedding might be an over-smoothed representation that merges node features of different labels.

The interaction between the graph structure and the model performance has been investigated in several works. One question of interest here is how and under which circumstances the graph structure hinders or benefits graph deep learning. In this context, several graph metrics that assess the graph structure have been introduced that are strongly correlated with GNN performances. One such metric is *homophily* (and its counterpart: heterophily) (Luan et al., 2021; Ma et al., 2022), which quantifies the similarity of neighbouring labels. Originally, GNNs were built on the assumption that connected nodes share similar properties (they are homophilic), and GNNs perform well based on this assumption (McPherson et al., 2001). As a result, several graph DL models underperform on datasets with diverse neighbourhoods (heterophilic graphs). More metrics and their impact on GNN performance are discussed in Section 5.2.

Even though this line of research implies that the graph structure and, therefore, the graph construction method strongly impacts the performance of GNNs, the analysis of the works of GNNs in medicine shows that there is no unique, clearly defined method nor any guidelines for creating the graph structures from the wide variety of medical datasets. In this work, we, therefore, survey recent works that address graph creation methods for graph deep learning tasks with a focus on medical data. The methods summarised in this review are not limited to applications on medical datasets, and we provide links to other non-medical domains in Section 6.

## 1.2 Contributions and outline

This work provides an overview of graph construction methods in medicine. We performed a literature search on *Google Scholar* based on keywords like "geometric deep learning", "graph neural network", "medicine", "population graph", "disease", "graph construction", and combinations of them. We summarise 78 works and categorise them by their graph construction method. The outline of this work can be summarised as follows:

- We identify three types of graphs that can be distinguished: population-level graphs, subject-level graphs, and subject-independent graphs, which we use to categorise the included works (Section 3), as well as two structure types: relationship-based structures and spatially motivated graphs;

- In Section 4, we systematise existing works that utilise GNNs in medical application areas by graph construction methods with a focus on static and adaptive graph construction (see Figure 1);

- We formulate recommendations for choosing suitable graph construction methods in Section 5.3;

- We summarise existing graph assessment metrics that allow the evaluation of generated graph structures in Section 5.2;

- We embed our work into the context of related review papers in Section 6;

- We identify open challenges of graph learning in medicine (Section 7) and conclude with promising future directions of research (Section 8).

## 2 Background

In this section, we give an overview of graphs, GNNs, and homophily - a main graph property linked to the performance of GNNs.

### 2.1 Formal definition of graphs

Throughout this work, we discuss datasets involving graph structures. A graph $G := (V, E)$ is defined as a collection including a set of nodes/vertices $V$ and a set of edges $E$ connecting nodes. $n = |V|$ denotes the number of nodes in the graph. An edge $e_{ij} = (v_i, v_j)$ defines the connection from node $v_i$ to node $v_j$. A graph $G$ is undirected if and only if $e_{ij} \Rightarrow e_{ji}$, $\forall i, j \in \{1, \ldots, n\}$. All edges $E$ can be represented in the adjacency matrix $\mathbf{A}$ of size $n \times n$, where $\mathbf{A}_{ij} = 1$ if $e_{ij} \in E$ and 0 otherwise. A weighted graph $G_w := (V, E, \mathbf{W})$ additionally requires a weight matrix $\mathbf{W}$ that assigns a weight to every edge in the adjacency matrix. The weight matrix has the same dimensions as the adjacency matrix. A neighbourhood $\mathcal{N}_v$ of a node $v \in V$ is defined by a set of all nodes that have an incoming edge to node $v$: $\mathcal{N}_v := \{u \in V | e_{uv} \in E\}$. A node $v$ can be represented by a feature vector $x_v \in \mathbb{R}^m$. The features of all nodes (node features) can be summarised by the feature matrix $\mathbf{X} \in \mathbb{R}^{n \times m}$. In this work, we summarise, categorise, and investigate different methods to build the node feature matrix $\mathbf{X}$, the adjacency matrix $\mathbf{A}$, and the weight matrix $\mathbf{W}$ from different datasets.

### 2.2 Graph neural networks

Graph neural networks (GNNs) were first introduced by Gori et al. (2005) and further extended by Scarselli et al. (2008). The term summarises a branch of research that expanded DL methods to non-Euclidean datasets, using graph convolutions. Over the last years, several different graph convolutions have been introduced. GNNs are based on a message-passing scheme, where the information stored in the nodes is propagated among neighbouring nodes, following the graph's edges. We here define the message passing at the example of graph convolutional networks (GCNs) (Kipf & Welling, 2016) but note that the principle is easily transferable to other graph convolutions.

**Definition 2.1 (Graph convolutional networks (Kipf & Welling, 2016))** *Let $h_v^{(k)}$ define the feature representation of node $v$ at layer $k$. For GCNs, the initial node representation for all nodes $v \in V$ is defined as following:*

$$h_v^{(0)} = x_v. \tag{1}$$

*The node embedding of node $v$ at step $k$ is then defined as:*

$$h_v^{(k)} = f^{(k)} \left( W^{(k)} \cdot \frac{\sum_{u \in \mathcal{N}_v} h_u^{k-1}}{|\mathcal{N}_v|} + B^{(k)} * h_v^{(k-1)} \right), \tag{2}$$

*where the function $f$, the weight matrix $W$, and the bias $B$ are $k$-dependent learnable parameters that are shared across all nodes (Daigavane et al., 2021).*

The embedding of node $v$ of the previous step $(h_v^{(k-1)})$, as well as the sum of all neighbouring node embeddings, are combined in the new node feature representation at step $k$. Different graph convolutions use varied versions of this definition but also follow the message-passing scheme. For more information about GNNs, we refer to Wu et al. (2020).

### 2.3 Graph homophily and heterophily

A key statistical property of graphs that indicates how nodes of different labels are connected throughout the entire graph structure is homophily (Luan et al., 2021). This property has been shown to potentially have a significant impact on the performance of GNNs (Zhu et al., 2020). In general, three types of homophily can be distinguished that all focus on a different nuance of the metric: node, edge, and label homophily (Luan et al., 2021). *Edge* homophily (Zhu et al., 2020; Ma et al., 2022) in graphs is defined as the ratio of edges that connect nodes with the same label vs. different labels (see Equation 3). *Node* homophily (Pei et al., 2020) describes the average number of direct neighbours with the same label. *Class* homophily (introduced by Lim et al. (2021) and termed by Luan et al. (2021)) is an extension of edge homophily with the additional consideration for class imbalance. Formal definitions of node and class homophily can be found in Luan et al. (2021). We here define edge homophily since this is the most commonly used metric to assess graphs.

**Definition 2.2 (Edge homophily)** *Formally, the edge homophily of a graph $G := (V, E)$ and the set of node labels $Y := \{y_u; u \in V\}$ is defined as:*

$$h(G, Y) := \frac{1}{|E|} \sum_{e_{uv} \in E} \mathbb{I}(y_u = y_v), \tag{3}$$

*where $\mathbb{I}$ is the indicator function.*

In case half of the edges in a graph connect nodes with different labels and the other half connects nodes with the same label, the graph has edge homophily of 0.5. A graph is described as *homophilous* when $h$ is large (typically larger than 0.5) and as *heterophilous* otherwise (Kim & Oh, 2021). Homophily is only one metric to assess a graph structure and still holds some drawbacks regarding comparability between datasets and the direct impact on the performance of the downstream model (Platonov et al., 2022). More details about homophily and further graph assessment metrics can be found in Section 5.2.

## 3 Graph structures in medicine

Medical research and data often contain patient data that defines the structure of the dataset. We identify three distinct graph types that are used in medical applications: (1) *population-level graphs*, where typically individuals of a cohort are connected in a large graph, (2) *subject-level graphs*, where each subject is represented by an individual graph –leading to a multi-graph dataset–, and (3) *subject-independent graphs*, which represent more general structures, such as knowledge graphs, molecules or maps. Each graph type comes with individual challenges and utilises different methods for graph creation. In this section, we give an overview of those three graph types, which are visualised in Figure 2. We furthermore distinguish between two types of structures: (a) relationship-based and (b) spatial structures. Relationship-based structures use concepts and relationships to determine the graph structure and spatial structures use spatial information, for example, image key-points in Euclidean space. All graph types can be combined with all structure types. We summarise the combinations of graph types and structure types with examples in Figure 3.

### 3.1 Population graphs

One research area of graph learning in medicine utilises so-called population graphs (Figure 2(a)). They are generated by connecting all subjects in a cohort to a single (usually large) graph. The goal is to improve model performance by using interactions between the subjects/nodes in the graph. The most common structure of a population graph is one where every subject in the dataset is represented as a node and node connectivity is, for example, defined by some distance metric between the subjects. When using population graphs, the learning task of the GNN is usually *node prediction*. Here, a prediction (e.g. classification or regression) is made for every node. This can, e.g. be a disease prediction (Parisot et al., 2017) or age prediction (Kazi et al., 2022). Population graphs are an effective method to integrate multi-modal data and enable the usage of patient data from different data sources and modalities. There are some examples where population graphs are extended with some additional components. In Gao et al. (2021) e.g., the authors create a bipartite graph, where subjects and gene expressions are represented by node entities.

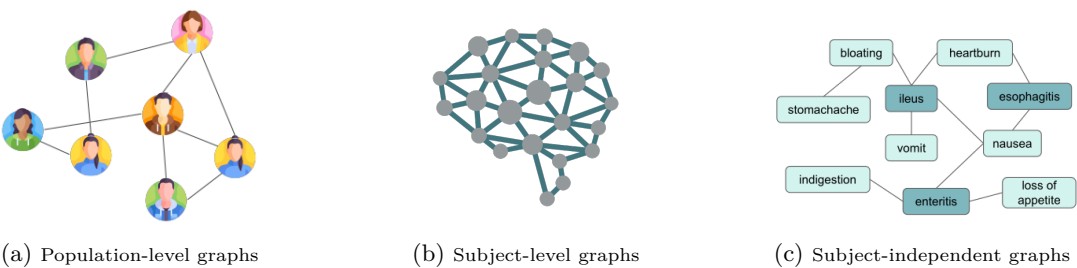

(a) Population-level graphs        (b) Subject-level graphs        (c) Subject-independent graphs

Figure 2: **Schematic display of the three graph types** utilised in medical research with GNNs: **(a)** population-level graphs, where each node represents a subject and subjects are connected based on similarity, **(b)** subject-level graphs with the example of a brain connectivity graph, where each subject is represented by a separate graph structure **(c)** subject-independent graph structures, here represented by a schematic display of a knowledge graph. The last category summarises graphs that represent more general aspects compared to subject- or population-level graphs that are not linked to medical subjects.

### 3.2 Subject-level graphs

Another way to represent medical data is in the form of *subject-level* graphs. This term summarises various graphs, where each subject in a dataset is represented by a single graph. The individual graphs are therefore independent of each other and together constitute a multi-graph dataset. One commonly used example for a subject-level graph is the representation of brain images as a brain connectivity graph (see Figure 2 (b)). In general, there are numerous ways to create subject-level graphs, depending on the dataset at hand and the application. They can, for instance, be used to represent structural connectivity in graph representations of arteries (Chen et al., 2020b), brain vessels (Paetzold et al., 2021), or airways (Zhao & Yin, 2021; Selvan et al., 2020), or to model a skeleton of a human in motion (He et al., 2022). When using subject-level graphs, the most common learning task is *graph prediction* (e.g. classification or regression). However, there are also applications where node-level and edge-level predictions (Chen et al., 2020b) are targeted using subject-level graphs.

### 3.3 Subject-independent graphs

As a third category, we summarise subject-independent graph structures that represent more general concepts and data. The graphs in this category represent structures that are independent of individuals, such as molecules or maps, that are not tailored to subjects or cohorts of patients – in contrast to subject- or population-level graphs. This includes knowledge graphs, which encode general concepts and knowledge in graph structure, highlighting relations between different entities. They are often utilised in the context of diseases, symptoms, drugs, or genes to display their correlation or interaction. They are usually not personalised but are "intended to accumulate and convey knowledge of the natural world, whose nodes represent entities of interest and whose edges represent potentially different relations between these entities" (Hogan et al., 2021). An example of a multi-modal medical knowledge graph is PrimeKG (Chandak et al., 2023), which includes information about drugs, diseases, phenotypes, exposures, and genes. Cheng et al. (2021a) construct a knowledge graph on stroke data, and Bonner et al. (2022) review different knowledge graphs on biomedical data for drug discovery. A knowledge graph differs from population graphs in the sense that here, no individual patient data is represented as a graph, but general knowledge and connections between entities are modelled in graph form. For a more detailed review of knowledge graphs, we refer to Ye et al. (2022). Knowledge graphs can be used for different applications, either on their own or as an additional source of information for other tasks, like in Pfeifer et al. (2022). Another example would be the encoding of cartographical proximity in maps like connecting hospitals in different regions of the country (Jin et al., 2021) or connecting cities based on their local proximity (Yu et al., 2021c) or molecules, where nodes represent atoms and the edges bindings between them (Bonner et al., 2022). We consider molecules as another example of subject-independent graphs. Molecule-based datasets in the medical domain are commonly used for graph-level predictions, for example investigating drug properties (Duvenaud et al., 2015; Kearnes et al., 2016) or potential interactions between different drugs (Xu et al., 2019).

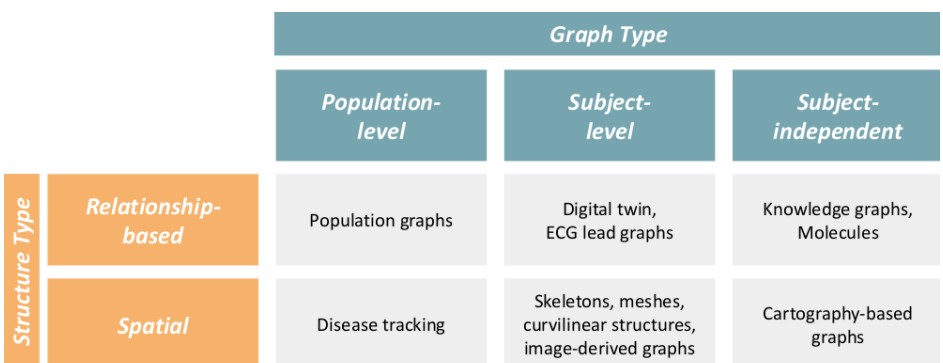

Figure 3: **Categorisation of graph types and structure types** with examples for the different categories. We consider three graph types: *population-level*, *subject-level*, and *subject-independent* graphs, which represent overall structures that are not linked to medical subjects. Additionally, we distinguish two types of structures: ones where the edges are based on relationships between nodes and ones that follow spatial information.

## 4 Graph construction methods

In the following, we summarise and categorise the state-of-the-art graph construction methods for medical data, linking them to the different graph and structure types introduced in Section 3. For graph deep learning, graph structures are usually extended to contain node features. This additional knowledge is then propagated along the graph structure during the learning process.

Generally, the graph construction process consists of two aspects: (1) defining the nodes and their features and (2) defining connections between the nodes (edge construction). Each node has a feature vector of shape $m$ and all node features of a graph $\mathcal{G}$ with $n$ nodes are summarised by a feature matrix $\mathbf{X} \in \mathbb{R}^{n \times m}$. The edges can be summarised by the adjacency matrix $\mathbf{A} \in \mathbb{R}^{n \times n}$. The definition of the adjacency matrix and node feature matrix are in general intertwined, and both steps are necessary to extract the full graph structure.

In the following sections, we investigate the definition of nodes and node features as well as the creation of the graph structure itself (the edges). An overview of the categories for graph construction methods, with a focus on the definition of the graph's edges, is visualised in Figure 4.

### 4.1 Defining the graph's nodes

The extraction of the graph's nodes is highly dependent on the constructed graph type. When building **population-level graphs**, every node usually represents a subject in the dataset. Node features can, for example, contain tabular data like lab results (Parisot et al., 2017), images (Keicher et al., 2021), image-derived features (Parisot et al., 2017), or combinations of those (Keicher et al., 2021).

For **subject-level graphs**, the node feature extraction can vary greatly. We here summarise the most prominent node and node feature definition strategies for different subject-level graphs. For the creation of brain connectivity graphs from fMRI data, the most commonly used approach to define node features is to define regions of interest (ROIs). Here, the definition of nodes is often guided by a 3D atlas, which defines the ROIs from the recorded BOLD signal (Wang et al., 2022b). We, therefore, say the definition of the graph nodes relies on prior knowledge (it is prior-driven). There are some examples where slightly different approaches are utilised to define the final nodes of a graph connectivity graph. For example, Zheng et al. (2022a) identify the brain's most informative regions through sub-graph generation. Yao et al. (2021) use several templates with varying ROI parcellation scales to create coarse-to-fine brain connectivity networks for each subject instead of depending on a specific brain parcellation. Yao et al. (2022), for instance, build a graph from image data and use features extracted from a convolutional neural network (CNN) as node features of the graph. To extract a subject-level graph that represents curvilinear structures, like vessels or airways, branching points of the structure can be used to define the nodes of the graph. However, nodes

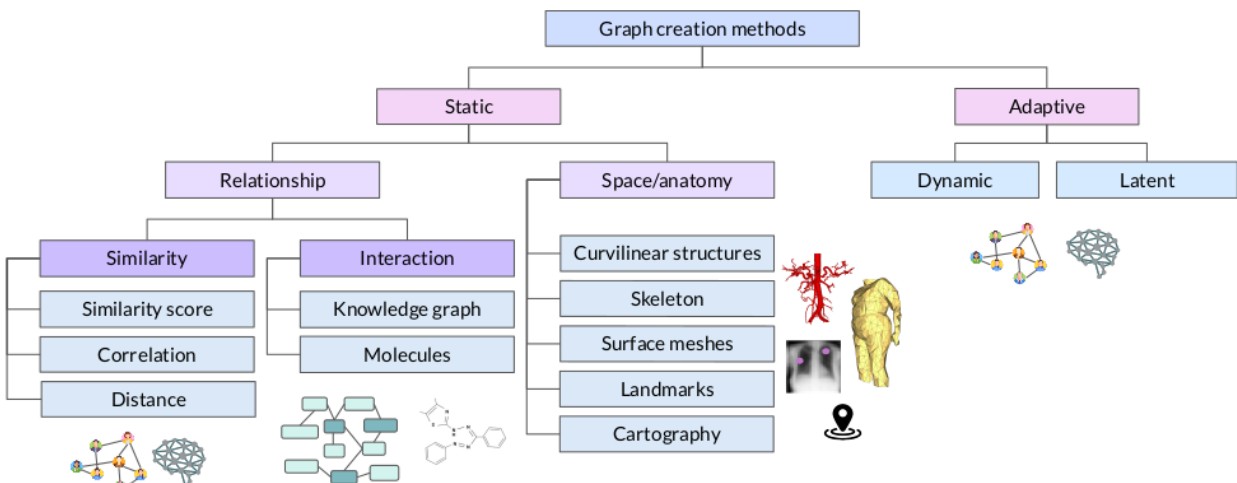

Figure 4: **Overview of graph construction methods** for GNN training in medicine categorised by static and adaptive graph construction. The icons represent examples of represented data in the respective categories. For static graph construction methods, the previously introduced separation of structure types by relationship-based and spatial interactions remains.

can also represent a section of the tube (Paetzold et al., 2021). Since in most cases, the definition of node features and edges for subject-level graphs are intertwined, we provide more information about the node features of subject-level graphs in Section 4.2.

**Subject-independent graphs** cover a wide range of different graph structures and datasets that all represent data that is independent of patients or subjects. This includes knowledge graphs, molecules, or cartography-based data of maps. When building knowledge graphs, the nodes typically represent entities of interest, such as diseases, symptoms, or medications. For molecules in drug research, for example, the nodes usually represent the molecules' atoms (Zheng et al., 2021; Zhao et al., 2021), and for cartography-based graphs, cities or hospitals can, for instance, be used to encode the graph's nodes (Yu et al., 2021c; Jin et al., 2021).

## 4.2 Defining the graph's edges

In this section, we introduce different methods for defining the structure of the graph (the edges). We hereby categorise existing works based on the following criteria:

(a) Graph type: population-level, subject-level, and subject-independent graph structures

(b) Static and adaptive graph construction mechanisms

(c) Purely data-driven and prior-driven methods

Graph creation methods can be categorised into *static* and *adaptive* approaches. We consider a graph creation method as *static* if the adjacency matrix is generated prior to training without any adaptions during the learning pipeline and as *adaptive* if the graph structure is adapted during training. The different static and adaptive graph creation methods are summarised in Table 1 and 2, respectively. Both tables also indicate the generated graph type and whether the approach is *data-* or *prior-driven*. We consider methods that only use the dataset at hand as *data-driven* and ones that also include additional prior knowledge as *prior-driven*.

## 4.3 Static graph construction methods

In this section, we categorise different static graph construction methods by the metric and information utilised to define the edges between the nodes. We mainly discuss graph type, connection mechanism, and

the utilised data. The available methods are summarised in Table 1. In column "Category" we categorise each method by the utilised relation for edge construction. "Graph type" indicates the specific type of graph (Section 3) that is generated, "Graph construction" refers to the utilised metric or property to decide when an edge is added to the graph, the represented data is listed in the respective column, "P/D" indicates whether the graph is prior- or data-driven and the listed references give examples of works that utilise the respective methods, including the initial introduction of a specific method.

Table 1: **Summary of static graph construction methods**. We differentiate between different graph types, construction methods, and source data types. We also indicate if the method is prior-driven (P), data-driven (D) or both (P, D), as well as if the graph is weighted (**W**).

| Category | Graph type | Graph construction | Represented data | P/D | W | References |
|---|---|---|---|---|---|---|
| Similarity and distance | Population level | Similarity score | Patient cohort | D | ✗ | Parisot et al. (2017); Ghorbani et al. (2022); Vivar et al. (2021) Pellegrini et al. (2022); Peng et al. (2022); Pan et al. (2021); Kazi et al. (2019); Ghorbani et al. (2021) |
| | Population level | Similarity score | Image features, clinical data | D | ✓ | Lin et al. (2023); Qiu et al. (2021) |
| | Population level | Mutual information | Images, clinical data | D | ✗ | Keicher et al. (2021) |
| | Population level | Euclidean distance | Image features, clinical data | D | ✗ | Lu et al. (2022); Yu et al. (2021b) |
| | Subject level | Euclidean distance | EEG signal | D | ✗ | Demir et al. (2021) |
| | Subject level | Euclidean distance | Images | D | ✗ | Sun et al. (2021) |
| | Subject level | Euclidean distance | Airways | D | ✗ | Tan et al. (2021) |
| | Subject level | Cosine similarity | Images | D | ✗ | Mahapatra et al. (2022) |
| | Subject level | Morphological similarity | fMRI data | D | ✗ | Mahjoub et al. (2018) |
| | Subject level | Pearson correlation | Brain connectivity | D | ✗ | Kim et al. (2021) |
| | Subject level | Partial correlation | Brain connectivity | D | ✓ | Li et al. (2021c) |
| | Subject-independent | Conditional probability | Lesion types | D | ✗ | Cheng et al. (2021b) |
| Relation | Population level | Medical assessments | Patient cohort | D | ✓ | Mao et al. (2022) |
| | Subject-independent | Known interactions | Disease/symptom/medication | P | ✓ | Ruan et al. (2021) |
| | Subject-independent | Interactions | Ontologies | P | ✓ | Hao et al. (2021) |
| | Subject-independent | Synergism/antagonism | Drugs | P | ✓ | Zheng et al. (2021) |
| | Subject-independent | Drug–protein pairs | Drugs | P | ✓ | Zhao et al. (2021) |
| | Subject-independent | Protein interactions | Proteins | P | ✗ | Schulte-Sasse et al. (2021) |
| | Subject-independent | Co-occurrences | (Clinical) abnormalities | P | ✓ | Liu et al. (2021); Zhou et al. (2021) |
| | Subject-independent | Co-occurrences | Medical labels | D | ✓ | Hou et al. (2021) |
| | Subject level | Medical importance | ECG leads | P | ✗ | Mueller et al. (2022a) |
| | Subject level | Protein interactions | Proteins | P | ✓ | Pfeifer et al. (2022) |
| Space, structure, or anatomy | Subject level | Local proximity | Image landmarks | D | ✗ | Yao et al. (2022); Hansen & Heinrich (2021); Huang et al. (2023) |
| | Subject level | Tree structure | Curvilinear structures | D | ✓ | Yu et al. (2022a) |
| | Subject level | Tree structure | Curvilinear structures | D | ✗ | Paetzold et al. (2021); Wittmann et al. (2023); Shin et al. (2019) Chen et al. (2020b); Wolterink et al. (2019); Yu et al. (2022b) |
| | Subject level | Local proximity | Curvilinear structures | D | ✗ | Xu et al. (2022); Xie et al. (2022) |
| | Subject level | Local proximity | Curvilinear structures | D | ✓ | Li et al. (2021b) |
| | Subject level | Anatomy | Variety of patient data | D | ✗ | Barbiero et al. (2021) |
| | Subject level | Anatomy | 3D point clouds | D | ✗ | Yu et al. (2021a) |
| | Subject level | Anatomy | Skeleton | P,D | ✗ | He et al. (2022); Deb et al. (2022) |
| | Subject level | Anatomy | Curvilinear structures | D | ✗ | Selvan et al. (2020) |
| | Subject level | Mesh generation | Cortical surface meshes | D | ✗ | Azcona et al. (2020); Gopinath et al. (2019a); Wu et al. (2019); Gopinath et al. (2019b) |
| | Subject-independent | Cartography | Maps | D | ✗ | Yu et al. (2021c); Jin et al. (2021) |
| | Subject-independent | Chemical structure | Molecules | D | ✗ | Bonner et al. (2022); Kearnes et al. (2016); Duvenaud et al. (2015) |
| | Subject-independent | Text-derived | Disease/symptom/medication | P,D | ✓ | Vretinaris et al. (2021) |
| | Subject-independent | Chemical structure | Disease/symptom/medication | P,D | ✓ | Zhang et al. (2022b) |

### 4.3.1 Graph construction based on relations

Following the in Section 3 introduced structure types (relationship-based and spatially motivated structures), we can differentiate these two types of static graph construction. Relationship-based graph construction methods define a relation between node features, such as similarity or interactions, and spatially motivated graph construction methods utilise spatial information to define edges.

**Similarity-based graph construction**

One method that is frequently used to determine whether nodes connect is based on a similarity or distance measure. Simplified speaking, in this approach, two nodes will be connected, if they are "similar" or "similar enough". The similarity/distance between nodes can be defined by different metrics, including a similarity score, the Euclidean distance, cosine similarity, or correlation-based similarity. Furthermore, methods using similarity or distance measures to construct the graph structure can be categorised into ones using the original data at hand, or a (mostly lower-dimensional) embedding of the original data (Lu et al., 2022). When using a similarity or distance measure to determine whether nodes should be connected, usually an additional step of edge selection needs to be performed to obtain the final graph (Section 4.3.4).

**Similarity score**  The similarity between nodes in the graph can be defined with the help of a similarity score. This is one of the most commonly used methods for the creation of population graphs in order to determine how similar two subjects in a cohort are. This method was first introduced by Parisot et al. (2017) in 2017 and has since then been used in many variants and applications, e.g. in Ghorbani et al. (2022); Vivar et al. (2021); Pellegrini et al. (2022); Peng et al. (2022) and Lu et al. (2022). One advantage of this method is that the similarity score considers discrete and non-discrete features and can be adapted by adding additional weight factors. All available features are divided into two groups (e.g. imaging and demographic features), then the graph is built based on a similarity score that uses one of these subgroups of features (e.g. demographic features). The second group of features (e.g. imaging features) are used as node features. Parisot et al. (2017) originally used this approach for brain analysis in Alzheimer's disease and autism prediction. They used phenotypic information for the graph creation and imaging features as node features and show an example of multi-modal data integration.

Variations of this method have been used for bone age estimation (Du et al., 2015), brain age prediction (Stankevičiūtė et al., 2020), autism detection (Rakhimberdina et al., 2020), genome inference (Dilthey et al., 2015), disease prediction (Chen et al., 2020a; Kazi et al., 2019; Ghorbani et al., 2021), to only name a few. In each of the here mentioned works, the original approach of creating the population graph by Parisot et al. (2017) has been modified to best fit the data and the application. Some adaptations of this method have also used both imaging and non-imaging features for graph creation as well as node feature representation (Lin et al., 2023) to maximise the information about the similarity between individual subjects.

**Correlation-based similarity**  Another option to create the graph structure is by evaluating the correlation between node features and connecting those nodes that show high correlation. Similar to the similarity-based creation, the correlation is thresholded and only nodes that are "correlated enough" will be connected by edges (Section 4.3.4). This method is used mainly for the creation of brain connectivity graphs from functional magnetic resonance imaging (fMRI) data. Here the correlation between the blood oxygen level dependency (BOLD) signal, indicating alterations in blood oxygen levels over time of all previously defined regions of interest (ROIs) –which are represented by the nodes in the graph– is first averaged within each brain ROI. Then, a correlation metric is calculated between pairs of regions, and the nodes that show a high correlation are connected leading to functional connectivity that illustrates the communication between different brain regions.

Pearson's correlation coefficient, which measures the linear correlation between nodes, has been commonly used to evaluate functional connectivity. Values range from $-1$ to $1$ where the former shows perfect negative correlation and the latter perfect positive correlation. A value of $0$ indicates no correlation. Alternatively, other correlation methods such as partial correlation, Ledoit-Wolf (LDW) regularised shrinkage estimator (Noman et al., 2021) and Spearman's rank correlation (Yu et al., 2023) are utilised to construct functional connectivity. Partial correlation is an extension of the Pearson correlation coefficient, which measures the linear relationship between two variables. By removing the effects of the controlling variables, partial correlation helps to identify the unique relationship between the two variables of interest, providing a clearer understanding of the associations between them. LDW regularised shrinkage estimator is a statistical method designed to improve the estimation of covariance matrices when the number of observations –number of scans– is small compared to the number of variables –number of ROIs. Noman et al. (2021) use this method to obtain well-conditioned functional connectivity. Spearman's rank correlation is a non-parametric rank correlation

method between two nodes that can detect monotonic nonlinear relationships. Yu et al. (2023) propose a multi-graph attention network to use both Pearson's and Spearman's rank correlation measures. Li et al. (2021c) utilise ROI-aware GNN assuming more closely connected ROIs exert a greater effect on each other and apply node (ROI) pooling layer (R-pool) to retain the most representative ROIs while eliminating noisy nodes. Gharsallaoui et al. (2021) predict an affinity matrix of the target view from the source view. Klepl et al. (2022) compare different functional connectivity measures for Alzheimer's disease prediction from EEG data as well as different edge selection methods.

One potential shortcoming of the creation of brain connectivity graphs from the BOLD signal is that the signal and the correlation between ROIs change over time. Kim et al. (2021); Kong et al. (2021); Wang et al. (2022b) target this by creating several brain connectivity graphs for different time steps to allow a more dynamic graph structure over time. Kong et al. (2022), e.g., propose a multi-stage learning module to fully utilise features at different stages and use a deep auto-encoder to extract the graph structure.

**Distance measures and mutual information**   In line with the usage of a similarity score, the *Euclidean distance* or the *cosine similarity* has also been used to determine the "distance" between nodes  (Lu et al., 2022; Yu et al., 2021b).  These distance measures can also be interpreted as a similarity between nodes and used as a basis for edge selection (Section 4.3.4), where only the least distant nodes are connected. Keicher et al. (2021) use images, extracted image features, and clinical data as node features and build the graph based on *mutual information.*

### Interaction-based graph construction

Apart from similarity- and correlation-based edge definitions, the graph structure can also be derived from (usually known) interactions between entities. This can e.g. be symptoms that are related to certain diseases, medications that are used to treat diseases, or lab results that are associated with specific doctor's visits (Mao et al., 2022).

**Knowledge graphs**   Knowledge graphs are typically constructed based on known relations between entities that are usually extracted from large knowledge resources or clinical studies. Medical knowledge graphs can for example contain diseases, symptoms and medications, connecting co-occurring symptoms and diagnosed diseases with prescribed medication and reported symptoms. Even though the graph is constructed based on general knowledge, several design choices still remain. Zhang et al. (2020) create a knowledge graph based on clinical studies where abnormalities are connected with organs and body parts. Hou et al. (2021) follow the same graph creation approach, with an additional step of post-processing where missing links are included in a data-driven manner. Protein-protein interactions can be encoded in knowledge graphs and have, for example, been used to identify cancer genes  (Schulte-Sasse et al., 2021) and drug–target interactions (Zhao et al., 2021).

The knowledge graph PrimeKG  (Chandak et al., 2023) was for example built on large resources like *Drug-Bank*  (Assempour et al., 2018), *Drug central*  (Avram et al., 2021), and *Entrez gene*  (Maglott et al., 2010) - to mention only a few. The authors highlight that PrimeKG contains edges between drugs and diseases that specify "indications", "contradictions", and "off-label use", which are often missing in other medical knowledge graphs  (Chandak et al., 2023).

These constructed knowledge graphs can then be used as a basis for decision-making and graph learning. The authors in  (Vretinaris et al., 2021) for example use a knowledge graph to combine generally applicable information regarding medications and symptoms with text-derived patient-specific information about their individual symptoms. Zhang et al. (2022b) utilise a similar technique and utilise an attribute graph, containing chemical structures of drugs as an additional graph to the knowledge graph.

**Prior knowledge**   The graph structure can also be created or influenced by the inclusion of prior knowledge about the data and the importance of specific features in other areas apart from knowledge graphs. This was for example applied in a graph classification task using electrocardiogram (ECG) data to determine left bundle branch blocks  (Mueller et al., 2022a), where prior medical knowledge about the importance of seeds guided the graph construction method.

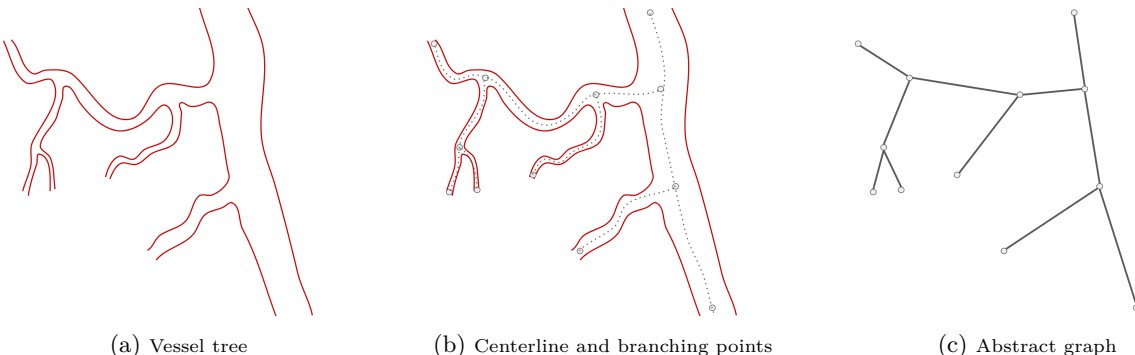

(a) Vessel tree          (b) Centerline and branching points          (c) Abstract graph

Figure 5: **Example of an abstract graph representation of a vessel tree**, with nodes encoding branching points and edges encoding vessel segments. Given a vessel segmentation extracted from a 3D or 2D image (a), centerlines and branching points are automatically or semi-automatically extracted (b), from which a graph is defined (c). Although the graph is inherent to the structure of the vessel tree, the extracted graph ultimately depends on the entire processing pipeline, which is inherently noisy.

**Pre-defined structural connections**    Some datasets come with internal structural information, from which the graph can be extracted. An example of this would be molecules, which have been used for drug discoveries  (Bonner et al., 2022) or protein analysis  (Pfeifer et al., 2022). Here, the graph creation usually follows the molecular bindings. For more information about these pre-defined structural connections, we refer the interested reader for example to a review on ML approaches for the design of multi-target drugs (Moreira-Filho et al., 2022).

### 4.3.2   Graph construction based on space, structure, or anatomy

Another graph construction technique is building spatio-structural or spatio-semantic graphs, where the nodes have associated positions in a metric space. In this case, the relations between the nodes usually correspond to distances in the respective metric space. The starting point can e.g. be a 2D or 3D image or a map, and a common choice for the metric space is the Euclidean space.

**Curvilinear structures**    One prominent application of spatio-structural graphs in medical imaging is represented by curvilinear structures. These are long, thin, structurally constrained tubular structures, following a tree or network configuration. Examples of such structures are vessel trees, airways, or neurons. As an exemplification of their structural constraints, vessel trees follow blood flow constraints, their branching pattern is dictated by the perfusion requirements of nearby tissue, and the cross-sectional diameter of constituent vessels progressively narrows in the distal direction. The inherent graph structure of these objects makes them ideal candidates for geometric deep learning: either by achieving a more compact, abstract representation for branch classification or exploiting geometric information for an auxiliary task. The graph representation is typically only a secondary representation derived from an initial imaging representation, making graph learning highly dependent on the quality of the graph extraction process. Figure 5 shows an example of a graph representation derived from a vessel tree.

The most straightforward representation of curvilinear trees as graphs follows the structure of the trees, where branching points form the node set and individual segments are edges. Such an approach is suitable for multi-class classification tasks, such as airway or vessel labelling. Chen et al. (2020b) construct a graph of the intracranial arteries by choosing nodes as centerline points with more than two neighbours and linking them based on skeleton connectivity. Their approach to multi-class vessel labelling is formulated as simultaneous node and edge classification, where message passing occurs through both node and edge representations. Many other works opt to formulate the branch labelling problem as node classification instead since node classification is more widely spread than edge classification. Paetzold et al. (2021) introduced the VesselGraph dataset of graphs derived from mouse brains for the tasks of link prediction and vessel classification. The graph structure is extracted by first generating a segmentation of a 3D volume, then processing it via the

graph extraction tool Voreen (Drees et al., 2021). Then the initial graphs with nodes as branching points and vessels as edges are subsequently converted via the line graph approach, such that nodes of the derived graph correspond to the edges of the initial graph. Xie et al. (2022) performed anatomical labelling of airways by directly constructing a graph where nodes correspond to branches and assigning edges based on the connectivity in the image segmentation map. Yu et al. (2022b) used a similar graph construction approach for airway labelling while adding additional hyperedges between the children of a parent, thereby using two information passing pathways.

Another prominent use of curvilinear structures in graph-based deep learning is to improve image segmentation. The construction of the graph is less abstract in such scenarios, instead following the voxel structure of the data representation. Nodes are no longer well-defined anatomical structures such as branches or branching points but tend to be pixels or superpixels, while edges follow their connectivity in the image. Xu et al. (2022) first partition the input image into superpixels, which become the nodes of the vessel graph. Edges are then only defined between nodes corresponding to neighbouring superpixels in the original image. The superpixel-derived graph is used for vessel segmentation refinement, by predicting superpixel vesselness as a binary node classification problem. Similarly, Shin et al. (2019) proposed to train a node classifier for segmentation refinement also using vesselness prediction. However, in their approach nodes are represented by points sampled equidistantly along the centerlines rather than superpixels, whereas edges are based on node connectivity or geodesic distances.

Some works (Yu et al., 2022a; Li et al., 2021b; Tan et al., 2021) only use GNNs during training for improved CNN feature representation for image segmentation, rather than as a standalone task. Yu et al. (2022a) partition the image into super-voxels and sample a node per super-voxel, relying on travel distance between nodes in the ground truth for determining edges, while Li et al. (2021b) rely on corner sampling following skeletonization for node creation and a similar approach for determining edges. Tan et al. (2021) perform airway segmentation via CNNs with concatenated GNN features by combining nodes around landmarks and nodes sampled at random lung locations. The adjacency matrix is created by connecting the $k$ nearest neighbours, in addition to connecting another node with the same predicted semantic class for each node to improve homogeneity.

Wolterink et al. (2019) use a different approach for vessel segmentation. Instead of working in the pixel space, they turn to the polar space centred around the vessel centerline and formulate the vessel segmentation problem as regression. For each cross-section of the centerline, a node for the centerline and 24 additional nodes around the centerline are created. Thus, the GNN is trained to regress the distance to the centerline. Each non-centerline node is connected to its corresponding centerline node, the corresponding nodes in adjacent cross-sections, as well as 2 neighbouring nodes. Additional connections are defined to establish a triangle mesh, whose vertex coordinates are determined by the regressed distances in the polar space.

**Skeleton representation** Another application that follows anatomical structures for graph creation, is the representation of the human skeleton in graph structure. Deb et al. (2022) use skeleton representation and GNNs for the assessment of physical rehabilitation exercises. Similar representations have been used for the detection of Parkinson's disease, where skeletons are extracted from video data of patients (He et al., 2022). The authors use two different methods for generating the graph structure: (1) connecting the joints of the skeleton locally, following the underlying skeleton (prior knowledge), or (2) globally, where every joint of the skeleton is connected to one node at the position of the neck. This shows that even when the data contains natural connectivity, there are multiple ways to extract a suitable graph structure for graph deep learning. The different representations are visualised in Figure 6.

**Surface Meshes** Meshes are a special type of graph structure, that can benefit from graph deep learning Bronstein et al. (2017). A mesh usually represents a 3D structure in space, where nodes are aligned around the surface of an object. Edges usually build triangular faces between neighbouring nodes, which is why they are often also referred to as *triangulated* meshes. Meshes have also been used for medical data representation, such as cortical structures (Azcona et al., 2020; Gopinath et al., 2019a; Wu et al., 2019; Gopinath et al., 2019b) in neuroscience or organ surfaces (Mueller et al., 2022b). Meshes can, for example, be extracted from segmentations of 3D images, using methods like marching cubes (Lorensen & Cline, 1998). Other works

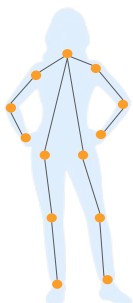 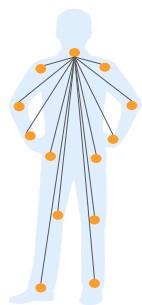 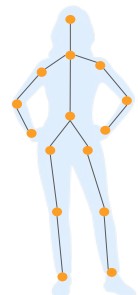

(a) Local connection module: here the graph structure follows the underlying skeleton of a person.

(b) Global connection module: all relevant joints are connected to one global node at the neck of a person.

(c) Another local connection module: here the spine is also included in the skeleton representation.

Figure 6: **Schematic visualisation** of two approaches to create a graph structure for skeleton representations that are introduced in He et al. (2022) and Duan et al. (2022).

utilise finite element meshes (Salehi & Giannacopoulos, 2022), for example, using methods like TetGen (Hang, 2015). Mueller et al. (2023c) use body surface meshes for the quantification of different types of fatty tissue. Surface meshes are frequently used in neuroscience. The authors in Salehi & Giannacopoulos (2022) use cortical meshes for soft tissue prediction in image-guided neurosurgery. Salehi & Giannacopoulos (2022), e.g., use meshes for predicting soft tissue deformation in image–guided neurosurgery. Azcona et al. (2020) use surface meshes for brain morphology estimation for Alzheimer's disease classification. They analyse cortical and subcortical irregularities that have been shown to be correlated with Alzheimer's disease. A review by Zhao et al. (2023) summarises methods for cortical surface-based neuroimage analyses, including methodologies for cortical surface extraction and parcellation. Popular software tools like FreeSurfer (Fischl, 2012) have integrated methods for cortical surface extraction from MR images.

**Image landmarks**  Graphs can also be extracted from images by identifying landmarks in the image and then connecting them based on local proximity. Hansen & Heinrich (2021) apply this method to extract lung landmarks from 3D computer tomography (CT) images in order to perform graph-based registration. They use the Foerstner interest operator (Förstner & Gülch, 1987) to identify sparse landmarks from the image, and then use a minimum spanning tree to construct the edges of the graph structure (Heinrich et al., 2015). Zhao (2021) use several graph structures in their work. One of them uses simple linear iterative clustering (SLIC) (Achanta et al., 2012) to determine the connection between super-pixels in chest X-ray images. Huang et al. (2023) use image patches as nodes, connect neighbouring nodes to construct a graph, and apply GNNs for image reconstruction.

**Cartography-based graphs**  There are a few works, where the graph structure follows cartographic proximity, like maps or regions. Jin et al. (2021) e.g. use hospitals and regions as nodes in their graphs, connecting geologically close entities. Yu et al. (2021c) suggest representing nodes by cities in the graph structure, and the proximity of geolocation is used to connect them.

### 4.3.3 Fully connected graphs

Similar to pre-defined graph structures, there are some applications, where a fully connected graph is used (Chao et al., 2020). In those scenarios, no graph creation method has to be chosen. Instead, all nodes are connected to all other nodes. Chao et al. (2020), for example, use fully connected graphs for lymph node gross tumour volume detection.

### 4.3.4 Edge selection

In many of the above-mentioned graph creation methods, a distance/similarity/correlation/proximity between nodes is derived. However, this does not directly lead to a final graph structure. In order to create the final (sparse) graph from the determined distance between the nodes, a set of –sometimes weighted–

edges needs to be selected. This is usually either done by (a) thresholding or (b) selecting a fixed number of neighbours (*k*-nearest neighbours).

**Thresholding edges** Thresholding techniques can be further split into *absolute* thresholding and *proportional* thresholding. In *absolute* thresholding, a fixed threshold is chosen to remove weak connections which results in varying graph densities for different subjects. In *proportional* thresholding, a portion of the strongest connections is kept resulting in all graphs having the same edge density. The latter method is for instance used by (Noman et al., 2021; ElGazzar et al., 2022). In addition to those methods, Yang et al. (2021) applied the spatial-constrained sparse representation optimisation method to obtain a sparse representation matrix that captures the relationships between brain regions while considering their spatial proximity.

**k-nearest neighbour selection** Alternatively, the *k*-nearest neighbours (*k*-NN) of each node can be selected based on the similarity/distance/proximity between the nodes. This way, every node is connected to a fixed number (*k*) of neighbours. *K*-NN graph creation methods are applied in several application areas, e.g. for population-level graph creation and brain graph creation. In the latter, a *k-NN graph* is derived from each densely connected functional connectivity matrix. It connects each brain region of interest to the *k* neighbours that show the highest connectivity. This is for example used in Zhang et al. (2021b); Qu et al. (2021); El Ouahidi et al. (2022); Wang et al. (2022a); Yao et al. (2021). For population-level graphs, usually, the *k*-nearest neighbours are selected based on the features extracted from the dataset (Lu et al., 2022; Yu et al., 2021b). An implementation of a *k*-NN graph creation is for example provided by PyTorch Geometric (Fey & Lenssen, 2019).

## 4.4 Adaptive graph construction methods

In contrast to the previously investigated static graph creation methods, we here introduce adaptive methods, where the graph structure is adapted during GNN training. The methods using adaptive graph construction are summarised in Table 2.

Zhu et al. (2021) propose a categorisation of adaptive graph creation methods into (a) metric-based, (b) neural-based, and (c) direct approaches. Metric-based approaches (a) use kernel functions to compute a similarity between node features or embeddings and use those to determine edge weights. Neural approaches (b) use neural networks to predict the edge weights, and direct approaches (c) interpret the adjacency matrix as free variables. They refer to adaptive graph creation methods in general as "graph structure learning" (GSL). However, these terms are not common in the medical domain, where instead the terms *dynamic* and *latent* are used. *Dynamic* approaches adapt the graph structure in between training steps, but do not specifically learn the graph structure in an end-to-end manner, which *latent* graph construction methods do.

Table 2: **Summary of adaptive graph creation methods** with their corresponding graph types and examples of original data, from which the graph is extracted. D indicates, that the graph creation method is data-driven and P that it is prior-driven.

| Category | Graph type | Method | Represented data | P/D | References |
|---|---|---|---|---|---|
| Dynamic | Subject-independent | Euclidean distance | Point clouds (non-medical) | D | Wang et al. (2019) |
| | Subject-independent | Euclidean distance | Proteins, molecules | D | Tran et al. (2018) |
| | Population level | Cosine distance | Patient cohort | D | Zheng et al. (2022b) |
| | Population level | Contrastive learning | Brain connectivity | D | Wang et al. (2022b) |
| | Subject level | Euclidean distance | Airways | D | Garcia-Uceda Juarez et al. (2019) |
| Latent | Population-level | End-to-end learned | Patient cohort | P,D/D | Kazi et al. (2022); Cosmo et al. (2020); Mullakaeva et al. (2022); Huang & Chung (2020); de Ocáriz Borde et al. (2023); Song et al. (2021) |
| | Subject level | End-to-end learned | Brain connectivity | D | Kan et al. (2022); Campbell et al. (2022); Mahmood et al. (2021); El-Gazzar et al. (2021); Zhu et al. (2022) |
| | Subject level | End-to-end learned | MR images | D | Mo et al. (2021a) |

### 4.4.1 Dynamic graph construction

Dynamic graph creation methods do not construct a fixed/static graph structure before training but change the graph structure depending on the embeddings during GNN training. Wang et al. (2019) propose dynamic graph CNNs (DGCNNs) for the analysis of point clouds. They extract a *k*-nearest neighbour graph based on the feature space of the applied neural network, which changes in between training steps. In Tran et al. (2018), the authors extend this method with a new definition of graph convolutional filters with the goal of improving the impact of distant neighbours. They use the shortest path connections and also do not optimise the latent space with respect to the graph structure but regarding the downstream task. Their *k*-nearest neighbour operation is not differentiable. Tran et al. (2018) also apply their method to biological graphs like molecules and proteins, however, they have not been applied to medical datasets specifically.

Zheng et al. (2022b) introduce a framework that utilises a dynamic graph generation method for disease prediction with a focus on the integration of multi-modal data. With the addition of separate loss terms, their method optimises for the downstream task and certain properties of the adjacency matrix of the graph, using attention. Wang et al. (2022b) use contrastive learning to generate a population-level graph from fMRI data using multiple views of the fMRI scans of every subject. They use partial correlations between ROIs to learn the graph structure that represents the fMRI data. Zhao et al. (2022) dynamically generate the graph structure by aggregating calculated *k*-NN graphs during training. This method retains more comprehensive non-local information diffusion compared to the proximity derived from a fixed input space.

Garcia-Uceda Juarez et al. (2019) use GNNs for improved CNN feature representation, by placing a GNN as the bottleneck layer inside a U-Net (Ronneberger et al., 2015). Features from the last encoder layer corresponding to encoded super-voxels serve as nodes connected either through regular grid connectivity, or based on the nearest neighbours in the feature space.

### 4.4.2 Latent graph learning

Here, we summarise methods that specifically learn the graph structure in an end-to-end manner. This requires the adjacency matrix of the graph to be differentiable. When using static graph construction methods, the adjacency matrix $A$ is often a binary discrete matrix, where an entry $a_{ij} = 1$ indicates that there is a connection from node $i$ to node $j$. When using latent graph creation methods, $A$ needs to be continuous, to allow backpropagation through the adjacency matrix (Figure 7). This is usually implemented by a *fully connected graph* with continuous edge weights that are updated during learning.

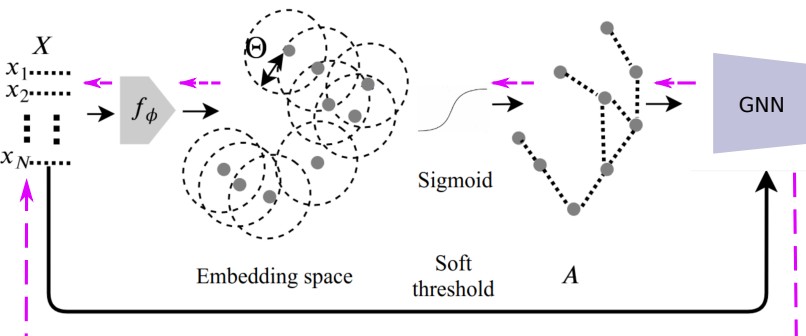

Figure 7: **Schematic overview of a latent graph learning approach**. Figure adapted from Cosmo et al. (2020); Kazi et al. (2022). The purple arrows indicate the backpropagation tracks through the networks. $f_\phi$ is an embedding network with learnable weights $\phi$ and A is the adjacency matrix, which will be updated.

The first works in this area used the spectral domain to learn the latent structure of the graph (Zhan et al., 2018; Li et al., 2018; Huang et al., 2018). These spectral methods require an initial graph which will then be changed over training and are limited to transductive learning settings since the extraction of the graph Laplacian requires the full graph including test and validation sets. Later, latent graph learning methods have also been transformed into the spatial domain, which also allows inductive training.

Cosmo et al. (2020) were the first to apply this approach in medical research for computer-aided diagnosis (CADx) and disease prediction. They could report improved performance when using a learned graph structure on two CADx problems (brain age prediction and Alzheimer's Disease prediction) compared to the approach of extracting a graph structure prior to training the GNN. In terms of graph structure learning, their method is classified as a "metric-based" GSL (Zhu et al., 2021). Mullakaeva et al. (2022) suggest extending the same graph learning method by adding additional loss to control the graph sparsity and applying it to a graph-in-graph learning task. Mo et al. (2021a) explore a similar technique for multi-modal data integration of MR images that also utilises an end-to-end trainable adaptive graph learning method. Huang & Chung (2020) also utilise an adaptive graph learning mechanism for disease prediction with the aid of population graphs. Their method includes a Monte-Carlo edge dropout for uncertainty estimation. This can be used to quantify the uncertainty of the edges learned in the graph creation method.

As an extension of Cosmo et al. (2020), Kazi et al. (2022) introduce the *differentiable graph module* (DGM). They propose two variants of their method with different sampling strategies: continuous and discrete. The latter has the advantage that a sparse graph can be sampled without the need for a fully connected adjacency matrix and requires the utilisation of the Gumbel-top-k trick (Kool et al., 2019) to ensure differentiability. In general, DGM can be applied based on an initial graph structure (prior-driven) or solely on the node features (data-driven). DGM uses Euclidean space to encode the latent features from which the latent graphs are decided. de Ocáriz Borde et al. (2023), extend the DGM to work on more complex embedding spaces by utilising Riemannian geometry.

One aspect in which static methods seem to show shortcomings in representing the full dimensionality of the data is when a temporal component is introduced. Functional brain connectivity networks e.g. fluctuate over time and can therefore benefit from an adaptive creation of the graph structure (Kim et al., 2021). We found five works that use similar approaches to learning the graph structure of functional brain connectivity networks end-to-end (Kan et al., 2022; Campbell et al., 2022; Mahmood et al., 2021; El-Gazzar et al., 2021; Zhu et al., 2022). In the method proposed by El-Gazzar et al. (2021), the adjacency matrix is randomly initialised, and its weights are adaptively learned along with the GNN weights by using gradient descent. Zhu et al. (2022) couple feature learning with dynamic graph learning into the GCN architecture. Campbell et al. (2022) and Mahmood et al. (2021) utilise self-attention in their model architecture. Kan et al. (2022) introduce loss components that enforce sparsity in the graph and maximise the similarity between adjacency matrices of the same group while minimising similarity between differently labelled subjects.

There are some methods for adaptive graph construction, that have –to the best of our knowledge– so far not been applied to medical data that we consider promising methods from which medical data analysis with GNNs could benefit. Some examples are pointer graph networks (Velickovic et al., 2020), dynamic graph message passing networks (Zhang et al., 2022a), and deep heterophily graph rewiring (Bi et al., 2022).

## 5 Discussion and recommendations

In this section, we give insights and recommendations for selecting suitable graph construction methods. We summarise the main advantages of static and adaptive graph creation methods (Table 3) and provide an overview of graph metrics, that can be used to assess the generated graph structure. We give recommendations regarding the selection of a graph type as well as the choice between static and adaptive graph construction and discuss some essential impacts of the graph structure on different graph convolutions, guiding a suitable choice of the latter for the downstream task.

### 5.1 Static vs. adaptive graph construction

Comparing Tables 1 and 2, we can see that fewer works utilise approaches that work with an adaptive graph structure creation, compared to the static approaches. In the following, we will compare both methods and discuss the shortcomings and benefits of both (Table 3).

Static graph creation methods are more computationally efficient during training since they do not require updating the graph structure. They require fewer trainable parameters and are easier to train in general. Backpropagating through the adjacency matrix and fine-tuning the adjacency matrix during training (adap-

Table 3: **Overview of advantages and disadvantages** of static and adaptive graph construction methods with respect to different aspects of the methods.

| Aspect | Static | Adaptive |
|---|---|---|
| Training efficiency | Computationally more efficient during training | No pre-processing w.r.t graph creation required |
| Complexity of training | Easier to train (fewer (hyper-)parameters) | More difficult to train |
| Final graph structure | Generally applicable for different problems | Adjacency matrix is fine-tuned to the problem |
| Further utility of graph structure | Application independent graph structure | Adjacency matrix usable for interpretability |
| Generalisability of method | Different datasets require different methods | General method, dataset independent |
| Prior knowledge | Can be easily included in the graph | Some methods allow inclusion of prior knowledge |

tive approach) adds additional complexity to the learning problem and can make training more difficult. On the other hand, the end-to-end learning of the graph structure eliminates critical design choices, since the adjacency matrix does not need to be defined prior to training.

Another advantage of adaptive graph creation methods is that the resulting adjacency matrix is fine-tuned to the specific task. This way, the most suitable adjacency matrix can be learned. This is not the case for static methods, where the edges are defined prior to learning. Whereas the latter makes the adjacency matrix more general and the same matrix can be used for different learning tasks on the same dataset.

Regarding the utility of the created graph structure beyond the specific downstream task, the adeptly generated adjacency matrix can also be used for interpretability purposes. Learned connections can indicate correlations between node components or highlight more important regions of the graph. Campbell et al. (2022) use their adaptive graph creation functional connectivity networks as a basis to identify sex-discriminate brain regions, which show higher importance within the decision-making progress of the algorithm.

## 5.2 Graph assessment

So far, we have investigated different methods to construct a graph structure from medical data and evaluated the advantages and disadvantages of different methods. Now we discuss methods that can be used to assess the constructed graph in terms of similarities in neighbourhoods. This can be done before or during model training, depending on the utilisation of static or dynamic methods. Several graph assessment metrics have been introduced to evaluate the impact of the graph structure on the learning of GNNs. Table 4 summarises the different metrics as well as a summary of their targets.

Table 4: **Summary of graph assessment metrics** that have been shown to be correlated to GNN performance and can be used to evaluate generated graph structures.

| Graph Assessment Metric | Reference | Details |
|---|---|---|
| Edge homophily | Zhu et al. (2020) | Ratio between edges connecting same and differently labelled nodes |
| Node homophily | Pei et al. (2020) | Average amount of neighbours with the same label |
| Class homophily | Lim et al. (2021) | Edge homophily with consideration for class imbalance |
| Adjusted homophily | Platonov et al. (2022) | Extension of edge homophily satisfying maximal agreement |
| Label informativeness (LI) | | Information quantity provided about a node's label |
| Probabilistic Bayes Error (PBE) | Luan et al. (2023) | Probability of a node being misclassified |
| Negative generalized Jeffreys divergence | | Analytic expression for PBE |
| Normalized total variation (NTV) | Luan et al. (2022) | Variation of the graph signal w.r.t. graph filters |
| Normalized smoothness value (NSV) | | Effect of the edge bias |
| Neighbourhood entropy | Xie et al. (2020) | Similarity of node embeddings within neighbourhoods |
| Center-neighbour similarity | | (Dis)similarity of the neighbours of a node |
| Aggregation similarity score | Luan et al. (2021) | Proportion of similarity weights after aggregation |
| Diversification distinguishability | | Proportion of nodes benefitting from diversification operations |
| Cross-class neighbourhood similarity (CCNS) | Ma et al. (2022) | Similarity measure for neighbourhoods |
| Label smoothness | Hou et al. (2019) | Dissimilarity of neighbouring labels |
| Feature smoothness | | Similarity between connected feature vectors |

Edge homophily (Zhu et al., 2020) (defined in Section 2.3) is one of the most commonly used metrics known to be correlated with the performance of GNNs. It has been shown that several graph convolutions do not perform well on heterophilic graph structures. However, homophily is not a *necessary* property for successful graph learning and heterophily affects different graph models at different rates (Zhu et al., 2020). Heterophilic graphs can have very different structures, containing either highly informative neighbourhoods or uninformative ones. These different notions of heterophily can be captured by other metrics, such as label informativeness (Platonov et al., 2022) or cross-class neighbourhood similarity (CCNS) (Ma et al., 2022). For that reason, a more general term *node distinguishability (ND)* has been used (Luan et al., 2023), which quantifies the difference in neighbourhoods more generally than heterophily/homophily.

Additional metrics have been introduced that have shown to impact the performance of GNNs (Luan et al., 2021; Ma et al., 2022; Luan et al., 2022). Luan et al. (2022) discuss in which settings GNNs are beneficial and when GNNs under-perform other ML models and introduce two metrics called normalized total variation (NTV) and normalized smoothness value (NSV) to measure the effect of edge bias. Xie et al. (2020) decide when neighbourhood aggregation may be unnecessary by evaluating the so-called neighbourhood entropy and centre-neighbour similarity in graphs. Luan et al. (2021) assess graph statistics with the aid of an aggregation similarity score and a diversification distinguishability metric, mainly capturing the linear relations of the aggregated node features. Ma et al. (2022) investigate the similarity in neighbourhoods of same-labelled nodes and call their metric cross-class neighbourhood similarity (CCNS). Mueller et al. (2023a) reduce the CCNS matrix to a single value they call CCNS distance. Label informativeness (LI) (Platonov et al., 2022) quantifies the information content a neighbour's label provides about the label of a node of interest and therefore gives more insights into different notions of heterophily. Luan et al. (2023) address the concept of node-distinguishability (ND), which is represented by several metrics (Ma et al., 2022; Luan et al., 2023). This concept is more general than homophily versus heterophily. Luan et al. (2023) also extend the idea of homophily by introducing probabilistic Bayes error (PBE) and negative generalised Jeffreys divergence as metrics. PBE describes the "probability of a node being misclassified when the true class probabilities given the predictors are known" (Luan et al., 2023) and the negative generalised Jeffreys divergence is an analytic expression for PBE. Hou et al. (2019) introduce two smoothness metrics: label and feature smoothness. Label smoothness is similar to the homophily metric and feature smoothness quantifies how similar the node features are between connected nodes.

There is no easy answer to what a "good" graph structure looks like. Some graph learning methods are highly sensitive to heterophilic graphs, while others work well on both homophilic and heterophilic graphs (Zhu et al., 2020). Different notions of heterophilic graphs (Platonov et al., 2022) show the complexity of assessing graph structures by their homophily value. We believe that the notion of node distinguishability (Luan et al., 2023) summarises the impact of graph characteristics on GNN performance best.

Given the complexity of assessing a graph structure, we advise using multiple graph assessment metrics – ideally before (and for adaptive methods also during) model training. They can shed light on the composition of the graph and can potentially reveal reasons for poor performance or guide a suitable graph convolution for performing the downstream task. The evaluation of different metrics that assess different qualities of the graph structure can be essential for understanding the complex interplay between graph structure and model performance (Platonov et al., 2022).

**Graph assessment beyond supervised node classification** While graph assessment metrics have mostly been applied for node-level classification tasks and under supervised training, they are not limited to these settings since they represent general graph properties and can equally be applied to graph-level or edge-level predictions. Homophily has initially been introduced for classification tasks and discrete adjacency matrices but has recently been extended to the notion of node homophily in regression tasks and continuous adjacency matrices, which are required for some adaptive graph learning methods (Mueller et al., 2023a). Furthermore, some metrics have been specifically used for link prediction and unsupervised learning tasks. Li et al. (2022) discuss unsupervised learning for node and edge classification tasks and link prediction, as well as the utilisation of edge labels for edge label assortativity. They observe that edge classification tasks rely more on features of paired nodes having signals with different frequencies. We notice a lack of more

detailed and specific graph assessment metrics for tasks beyond node-level classification and identify this as an open and important research area.

### 5.3 Selection of graph construction methods

We here formulate specific recommendations for selecting a graph construction method for different applications and datasets. We discuss the choice of graph type (population-level, subject-level, or subject-independent graphs) as well as the choice between static and adaptive graph construction.

**Choosing a graph type** The choice between the different graph types is highly dependent on the dataset as well as the desired downstream task. If a graph structure can be extracted for each subject individually, *subject-level graphs* are of use. Here, no dependencies between the subjects need to exist or be derived. *Population-level graphs* on the other hand aim to represent all subjects in one data structure. Each subject is represented by a feature vector or matrix and additional information is added by connecting the subjects. There are also works studying graph-in-graph problems, where subject-level graphs can be additionally connected to form a population-level graph (Mullakaeva et al., 2022). The authors show that nested graph approaches can improve performance. *Subject-independent graphs* are of interest, when no specific relation to a medical subject is available or relevant. This is for example the case when performing link prediction in knowledge graphs, where the relationships between entities (such as diseases and symptoms) are of interest and there is no personal data of patients.

**Choosing a static or adaptive graph construction method** The choice between a *static* or *adaptive* graph construction method is more challenging and depends both on the application and its goal. Knowledge graphs are typically created statically, since here, the connections between entities (e.g. symptoms and diseases) are based on literature and probably do not benefit from end-to-end learning. The same holds for molecules. Here, the graph structure is intrinsic to the data, and it would be unintuitive to change the edges (i.e. chemical bonds) of molecules. On the other hand, population-level graphs and some subject-level graphs have shown to benefit from an adaptive graph construction (Kazi et al., 2022; Campbell et al., 2022; Kan et al., 2022). They can, for example, be used to incorporate a temporal component into the learning process, like with fMRI data. It can also be a suitable method to altogether avoid the necessity of picking the *right* graph creation method before training since no specific metric or feature selection is required in this case. The main advantages and shortcomings of adaptive and static graph creation methods are summarised in Section 5.1. We believe that choosing an adaptive graph construction method is especially beneficial in settings like population graphs, where neither an initial graph structure nor a ground truth is available. We believe that this is a promising area for future research. Especially, the possibility of using the learned graphs to gain insight into the dataset at hand for interpretability purposes shows high potential for further investigation.

The most commonly used method to generate population-level graphs is based on similarity (Parisot et al., 2017) or distance (e.g. Euclidean) (Lu et al., 2022). In both settings, a subset of the available features can be used to define the edges of the graph (e.g. non-imaging features), while the remaining features (e.g. imaging features) can be used as node features (Parisot et al., 2017). This –in itself– implements an opportunity for multi-modal data integration. Later, it has also been shown that using all available features for both, creating the edges and the node features might be beneficial over splitting the features up (Keicher et al., 2021). The most frequently used approach for generating brain graphs is based on correlation, and graphs from curvilinear structures are mostly extracted based on the underlying tree structure of the vessels/airways.

Regardless of the graph construction method of choice, we highly recommend evaluating graph assessment metrics to gain insights into the graph structure and potentially evaluate whether it is beneficial to the learning task or might hinder GNN performance. This is especially useful when using static graph construction methods so that the graph structure can be judged prior to training.

## 5.4 Selection of Graph Learning Method

The graph construction method directly impacts the graph structure and therefore the performance of the utilised GNN for the downstream task. We, therefore, here discuss the interplay between the two components. The graph construction method can be either independent of the graph learning pipeline (static graph construction) or intertwined (adaptive graph construction). In both cases, the choice of graph convolution is an important factor. Zhu et al. (2020) identify some critical design choices that can improve the performance of GNNs on heterophilous graphs. They show that a separate embedding of neighbourhood node features and node-internal features improves the performance of heterophilous graphs as well as a separate embedding of higher-order neighbourhoods and introduces a new model architecture that works well on both homophilous and heterophilous graphs. Graph convolutions that propagate information simultaneously for all neighbours and their own node features are more impacted by heterophilic graphs than convolutions that have separate message-passing schemes for the neighbourhoods and their own node features. There are several graph convolutions, that have been specifically designed for low-homophily graphs, such as H2GCN (Zhu et al., 2020), HEAT convolutions (Mo et al., 2021b), or heterogeneous graph transformers (Hu et al., 2020b). In case the constructed graph results in a low-homophily graph, it is advisable to select one of these graph convolutions for graph learning. However, the choice of such graph convolutions does not guarantee high performance. We furthermore note that graph convolutions that are highly impacted by the graph structure (e.g. GCN (Kipf & Welling, 2016)) benefit most from the utilisation of adaptive graph construction methods. In these cases, the graph structure can be optimised for the whole graph learning pipeline, including the graph convolution at hand. Different models have also been specifically designed for specific tasks. RotatE (Sun et al., 2018), TransE (Bordes et al., 2013), and PairRE (Chao et al., 2021) have, for example, been designed for link prediction tasks and applied for knowledge graph completion. We furthermore identify a systematic evaluation of the connection between model architectures and graph structures as an open research question.

## 6 Related work

In this section, we put our work in context with existing research. Table 5 summarises adjacent works in similar fields and more information about GNNs and their application in different areas of medicine.

Table 5: **Summary of recent survey papers** in connected areas with their specific application areas, the year of publication, and the reviewed methods. *Categorisation methodology* lists the main categorisation schemes followed in the respective works. GSP refers to graph signal processing, and AE to autoencoders.

| Domain | Reference | Year | Reviewed methods | Categorisation methodology |
|---|---|---|---|---|
| Medical | Ahmedt-Aristizabal et al. (2021) | 2021 | GNNs | Model architecture, application |
| Medical imaging | Ding et al. (2022) | 2022 | GNNs | medical image type |
| Bioinformatics | Yi et al. (2022) | 2021 | Graph embedding, GNNs | Algorithm, application levels |
| Bioinformatics | Zhang et al. (2021a) | 2020 | GNNs | Application, prediction task |
| Neuroscience | Bessadok et al. (2022) | 2020 | GNNs | Application, loss function Architecture, graph creation |
| Biological data | Li et al. (2021a) | 2022 | GNNs, GSP | Application, feature extraction |
| Medical | Ours | 2023 | GNNs | Graph construction, graph type |
| Knowledge graphs | Ye et al. (2022) | 2022 | GNNs | Task, application |
| Recommender systems | Wu et al. (2022) | 2022 | GNNs | Type of information used |
| Fault analysis | Chen et al. (2021) | 2021 | GNNs | Application, data representation |
| Finance | Wang et al. (2021) | 2021 | GNNs | Graph construction, feature extraction |
| Text classification | Pham et al. (2022) | 2022 | GNNs, RNNs CNN, AE | Text to graph transformation, Model architectures |
| Domain invariant | Zhu et al. (2021) | 2022 | GNNs | Graph structure learning type |

**Reviews on GNNs on medical data** Ahmedt-Aristizabal et al. (Ahmedt-Aristizabal et al., 2021) give a detailed overview of GNN training on medical data, organised by application type and graph architectures. The authors investigate different graph creation methods, however, putting them mostly into context with varying application areas. They specifically name graph creation as one of the open challenges in research in this area. For an overview of works covering GNNs, different graph embedding techniques (homogeneous,

heterogeneous, and attributed), and generative graph models, we refer to Yi et al. (2022), who target works in bioinformatics. Zhang et al. (2021a) focus on GNNs in bioinformatics, and Bessadok et al. (2022) give an overview of GNNs in network neuroscience. The latter also categorise existing works by graph creation methods; however only refer to research in neuroscience, and summarise different loss functions for different applications. For a review of graph signal processing and graph learning on biological data, we refer to Li et al. (2021a). The authors also provide a summary of feature extraction techniques in the graph domain. Ding et al. (2022) give an overview of applications of GNN methods to multi-modal medical imaging datasets.

**Graph construction in other domains** Graph construction can be a challenging aspect of graph deep learning pipelines, regardless of the domain and application. Therefore, we list some works investigating graph creation methods in different domains.

For *recommender systems*, Wu et al. (2022) provide an extensive survey on GNN applications, including an elaboration of the graph creation methods in this area. However, they only focus on the post-processing of the graph structure before training since the initial graph structure is usually always provided by the bipartite user-item graph - which differs from several medical datasets. Han et al. (2022) highlight the difficulties of creating a graph for *retrosynthetic planning* with GNNs and propose a semi-dynamic approach for connecting different molecules with each other. Different graph construction methods for association graphs for *fault analysis* are discussed in Chen et al. (2021). Here, the authors categorise the graph construction methods into (a) using the k-nearest neighbour method, (b) using prior knowledge, and (c) using matrix completion. This categorisation is similar to the one we use in this review. However, we add the additional component of adaptive graph construction. For a review on GNNs in *finance*, including different graph construction methods, we refer to Wang et al. (2021). They distinguish three graph creation methods: data-based, knowledge-based, and similarity-based. This categorisation is similar to the one utilised in our work. However, we provide a more detailed methodological distinction between the graph creation methods. Pham et al. (2022) give an overview of graph creation methods for text analysis, categorising existing works based on text graphs and model architectures. In Zhu et al. (2021), the authors investigate general methods for graph structure learning, limited to the here termed "adaptive" graph creation methods.

## 7 Open challenges and future directions

In this section, we discuss open challenges that arise when working with GNNs in medicine that can be linked to graph construction methods and graph learning in general.

### 7.1 GNNs vs. graph agnostic models

As mentioned in Sections 1.1 and 5.2, recent works have shown that GNNs do not consistently outperform graph agnostic models (Luan et al., 2022). In most of the here summarised works, the original data does not come naturally in graph structure. Therefore, one could raise the question of when it is beneficial to construct a graph and perform graph learning at all. Most methodological works show that GNNs lead to improved performances compared to graph-agnostic baselines. However, sometimes GNNs show only little or no improvement over other ML methods. We see a detailed investigation of these connections to be a highly relevant area of future work.

### 7.2 Bias introduction

Another challenge that arises when choosing a graph creation method is the introduction of bias (Mehrabi et al., 2021). When creating a graph structure statically, every design choice (e.g. which features to select for graph creation and which to use as node features, the incorporation of prior knowledge) introduces (human) bias to the resulting graph structure. Sohn et al. (2015), for example, discuss the impact of the selection of ROIs for brain graph creation from fMRI data and argue that individual ROI selections would be more accurate. When using purely adaptive graph creation methods, no human bias will be introduced into the graph creation method. Here, the bias in the dataset will dominate the graph creation. If the available data

is highly imbalanced, questions of fairness might arise. This is something we believe should be considered when selecting a graph creation method, and we believe this to be an interesting area of research.

### 7.3 Undirected vs. directed graphs

Most approaches discussed in this work have been applied to undirected graphs, where edges are always bi-directional and only a few works discuss methods that work on directed graphs (Keicher et al., 2021). The effects of using directed versus undirected graphs have not been studied in detail. Changing an undirected graph to a directed one could be an extension of some of the here mentioned methods for graph construction.

### 7.4 Availability of public datasets

One significant challenge in exploring graph construction methods on medical data is the shortage of publicly available datasets. For population graphs, two commonly used public datasets are TADPOLE (Yu et al., 2020), a subset of the ADNI dataset, and the autism brain imaging data exchange (ABIDE) (Di Martino et al., 2014), which address Alzheimer's disease and autism detection, respectively. The private dataset UK Biobank (Sudlow et al., 2015) is also frequently used. The open graph benchmark (Hu et al., 2020a) holds a set of benchmark datasets for graph learning. It also contains molecule datasets and a vessel graph dataset from Paetzold et al. (2021). For the construction of brain connectivity graphs, the human connectome project (HCP) (Elam et al., 2021) and the brain connectivity challenge dataset[1] have been used apart from several private datasets. For surface mesh representations, MedShapeNet Scharinger et al. (2023) offers a variety of medical mesh structures.

### 7.5 Data privacy

Medical research is usually performed on private data such as medical images or patient records, which hold highly sensitive information about patients. A well-established, formal method to allow DL while giving privacy guarantees to individuals is Differential Privacy (DP) (Dwork et al., 2014). It ensures that the output of a randomised algorithm is approximately the same, independent of a single data point being in the dataset or not. It has been shown that guaranteeing DP in GNNs is more challenging than in tabular data since the individual data points (nodes) in a graph are inter-connected (Mueller et al., 2023b). Furthermore, graph-structured data is more sensitive to privacy attacks like Membership Inference Attacks (MIAs), which aim to identify whether a certain data point was part of the training dataset. This is because the additional information lies in the graph structure itself and can be leveraged by adversaries (Olatunji et al., 2021). We see a requirement for the development of high-utility privacy-preserving techniques for graph learning in medicine.

## 8 Conclusion

This in-depth review of state-of-the-art works on graph creation methods for medical data shows that graph construction is challenging and requires various design choices as well as a careful consideration of the dataset and task at hand. There are numerous ways to construct a suitable graph structure from a dataset and the best method needs to be selected with caution. We categorise graph construction methods by *static* or *adaptive* approaches. Static graph construction methods generate a graph structure prior to learning, whereas adaptive methods change the graph throughout GNN training. We analyse advantages and disadvantages of both approaches and formulate recommendations about how to pick a suitable graph construction method.

So far, adaptive methods have only been applied to a small subset of graph-learning tasks in medicine. We believe that this will be explored further in the coming years. Especially, the post-hoc interpretability of the learned adjacency matrix can hold valuable information about the dataset and task, which might not be possible to extract from a statically generated graph.

Given the findings about the strong impact of the graph structure on GNN performance (Section 1.1 and 5.2), we want to raise awareness that in cases where the graph structure is not clearly defined by the dataset,

---

[1]miccai.brainconnectivity.net

it is important to consider different graph creation methods as well as suitable graph convolutions. One characteristic of static graph creation methods is that the graph structure is defined before training. This opens up the possibility of evaluating the graph structure based on the metrics summarised in Section 5.2. We believe this to be a valuable step in evaluating the generated graph structure. This way, for example, the homophily of the graph can be analysed and the potential need for graph convolutions that can handle heterophilous graph structures can be evaluated (Du et al., 2022; Zhu et al., 2020). This could lead to graph learning methods that are more linked to the constructed/available graph structures. These graph metrics could also potentially be incorporated into the end-to-end learning of adaptive graph structures, where they could be added to the loss function to push the generated graph in the direction of a specific metric.

Given the various options to construct graphs from medical data, we think that explicit explanations of the applied graph creation method in each work would be beneficial to the research community. Even though the current state-of-the-art results summarised in this work indicate several recommendations for graph-creation methods on medical data, we believe that there are still a lot of open questions regarding which graph-creation method works best for which application. Furthermore, we consider a more thorough investigation of why a certain method might be beneficial over others to be necessary to learn more about the fundamental methodologies of GNNs and how they can be applied in the most favourable ways. We hope to contribute to further research in this area with this survey.

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
