# OpenReview forum: "A Survey on Graph Construction for Geometric Deep Learning in Medicine: Methods and Recommendations"
_TMLR — Accepted by TMLR_

### Review · Reviewer_UUrP · 2023-09-15

**Summary Of Contributions:**

This is a survey paper that reviews graph construction for geometric deep learning in medicine. The authors overview many relevant papers and discuss different types of tasks (graph level, node level, predicting relations), static and adaptive graph construction, and discuss current challenges and future research directions.

**Audience:**

Yes

**Broader Impact Concerns:**

Broader Impact is not discussed in the paper.

**Claims And Evidence:**

Yes

**Requested Changes:**

1. Please, provide a deeper discussion of homophily/heterophily measures, their drawbacks, and how they can be applied to different types of problems.
2. More explanation about the types of graphs considered in the paper would be helpful (see the comment above).
3. The paper is overall well-written, but there are some minor writing issues that I noticed:
- Abstract: footnotes are usually placed after punctuation marks;
- Section 2.1: "The Graph" -> "A graph";
- Citations can be used via \citet{}, e.g., in Section 3.3 this can fix "Cheng et al. (Cheng et al., 2021a)" (more examples in the text);
 - Section 7.4: "ADNI dataset ," (extra whitespace).

**Strengths And Weaknesses:**

First, I should note that I am not an expert in ML for medicine, thus I cannot assess the completeness and correctness of some parts of the submission.

Strengths:
- The paper covers a wide range of related papers
- The paper is well written

Weaknesses:
- My main concerns relate to the sections devoted to homophily/heterophily (Section 2.3 and Section 5.3). This subject is covered in this survey since homophily measures can be used to evaluate the usefulness of the constructed graph. However, most of the homophily measures are designed for node-level tasks and it can be non-trivial to apply them to graph-level tasks or to knowledge graphs. This subject is not covered in the text.
- Also, in [1] it was shown that popular homophily measures have certain drawbacks and thus cannot be used to compare homophily levels of different datasets. Moreover, it was shown that homophily is not a good measure to predict the performance of a graph neural network on a given dataset, and an alternative measure is proposed.

[1] Platonov O. et al. Characterizing Graph Datasets for Node Classification: Homophily–Heterophily Dichotomy and Beyond. arXiv preprint arXiv:2209.06177, 2022.
- The graph types are divided into Population graphs, Subject-level graphs, Knowledge graphs, and Real-world structures. The last category is not clear. For instance, can molecular graphs belong not to this category, but to subject-level?

---

> ### Author Response · Authors · 2023-10-20
> **Response to Review**
>
> We thank the reviewer for their remarks and input. We applied the mentioned changes regarding writing, typos, and citations. We specifically thank the reviewer for pointing us towards this relevant reference [1]. We added the additional metrics to the graph assessment metric section (5.2) and discussed them in more detail. We also use this work as a basis for more details about heterophilic graph structures and extended the summary of graph assessment metrics by the work from Luan et al. “When Do Graph Neural Networks Help with Node Classification Investigating the Homophily Principle on Node Distinguishability” and Hou et al. “Measuring and improving the use of graph information in graph neural networks.”
>
> We mention some of the shortcomings of homophily where we introduce the metric and refer to Section 5.2 for more details in Section 2:
> “Homophily is only one metric to assess a graph structure and still holds some drawbacks regarding comparability between datasets and the direct impact on the performance of the downstream model (Platonov et al., 2022). More details about homophily and further graph assessment metrics can be found in Section 5.2.”
>
> Additions to Section 5.2:
> “Heterophilic graphs can have very different structures, containing either highly informative
> neighbourhoods or uninformative ones. These different notions of heterophily can be captured by other metrics, such as label informativeness (Platonov et al., 2022) or cross-class neighbourhood similarity (CCNS) (Ma et al., 2022). For that reason, a more general term node distinguishability (ND) has been used (Luan et al., 2023), which quantifies the difference in neighbourhoods more generally than heterophily/homophily.”
>
> We added another paragraph to Section 5.2 discussing graph assessment metrics beyond node classification tasks. Given the limited utilisation of such metrics in broader application areas, we believe this to be an important area for future work.
>
> “Graph assessment beyond supervised node classification. While graph assessment metrics have mostly been applied for node-level classification tasks and under supervised training, they are not limited to these settings since they represent general graph properties and can equally be applied to graph-level or edge-level predictions. Homophily has initially been introduced for classification tasks and discrete adjacency matrices, but has recently been extended to the notion of node homophily in regression tasks and continuous adjacency matrices, which are required for some adaptive graph learning methods (Mueller et al., 2023a). Furthermore, some metrics have been specifically used for link prediction and unsupervised learning tasks. Li et al. (2022) discuss unsupervised learning for node and edge classification tasks and link prediction, as well as the utilisation of edge labels for edge label assortativity. They observe that edge classification tasks rely more on features of paired nodes having signals with different frequencies. We notice a lack of more detailed and specific graph assessment metrics for tasks beyond node-level classification and identify this as an open and important research area.”
>
> Based on the reviewers’ comments, we changed the categorisation of the graph types in our work, aiming to make the distinction between them more clear. We added an additional distinction between graph types (population-level, subject-level, and generic graph structures) and structure types (concept-based and spatial). We therefore adapted Figure 2, added Figure 3 to the manuscript, and adapted Figure 4 to match the new categorisation. We discuss the graph types and structure types the following way:
>
> Section 3:
> “Medical research and data often contain patient data that defines the structure of the dataset. We identify three distinct graph types that are used in medical applications: (1) population-level graphs, where typically individuals of a cohort are connected in a large graph, (2) subject-level graphs, where each subject is represented by an individual graph –leading to a multi-graph dataset–, and (3) subject-independent graphs, which represent more general structures, such as knowledge graphs, molecules or maps. Each graph type comes with individual challenges and utilises different methods for graph creation. In this section, we give an overview of those three graph types, which are visualised in Figure 2. We furthermore distinguish between two types of structures: (a) relationship-based and (b) spatial structures. Relationship-based structures use concepts and relationships to determine the graph structure and spatial structures use spatial information, for example, image key-points in Euclidean space. All graph types can be combined with all structure types. We summarise the combinations of graph types and structure types with examples in Figure 3.”

---

### Review · Reviewer_vdAj · 2023-09-28

**Summary Of Contributions:**

The authors present an overview of graph construction methods in the medical application domain for subsequent processing with artificial neural networks models. In addition to the systematisation of the construction methods the authors offer a discussion of the advantages and disadvantages of the approaches and a recommendation section.

**Audience:**

No

**Broader Impact Concerns:**

Not present.

**Claims And Evidence:**

No

**Requested Changes:**

The main utility of reviews is to collect a vast number of approaches and systematise them into a coherent hierarchical structure on the basis of few guiding principles and axis of variation. The utility of recommendations is then to guide practitioner choices. What seems to be missing in the current work is: 1) a more articulated hierarchical structure  and 2) an explicit connection between data/graph characteristics and design choices. Regarding the latter point, it would be of interest to study the relationship between the graph construction approaches and the resulting characteristics of the resulting graphs such as:  dimensionality of the attributes,  number of nodes, density of the edges, degree distribution, small-world properties, heterogeneity, graph curvature, graph spectrum, etc; and the downstream models' architecture choices. Empirical and theoretical investigations on these relations could form the basis of grounded recommendations.

As it stands, this review does not seem to be able to provide sufficient guidance.

**Strengths And Weaknesses:**

The main weaknesses of the paper are:
1. focussing only on the construction of the graphs but ignoring the coupling with the downstream models (i.e. certain constructions are useful for certain models, and not much can be said of one independently of the other)
2. the  systematisation of the construction methods is shallow, i.e. one additional level would be much more informative: the distinction in static and dynamic approaches should then be followed by articulating and abstracting the types of static and dynamic approaches
3. the recommendations are not justified and defended on the basis of any explicit empirical or theoretical basis, the authors simply say that <<Based on the state-of-the-art literature on graph learning in medicine, we here formulate some recommendations for selecting a graph construction method for different applications.>>
4. the recommendations regarding the type of graph constructions are often non informative: <<the choice between population-level, subject-level, knowledge graph, and graphs based on real-world structures to be determined by the application and the problem to solve>> or <<creating a graph based on real-world structures seems reasonable whenever there is such a defined structure at hand>> or <<choosing an adaptive graph construction method is especially beneficial in settings like population graphs, where neither an initial graph structure nor a ground truth is available>>
4. the application types are heavily biased towards very few specific cases, i.e.  brain imaging.

---

> ### Author Response · Authors · 2023-10-20
> **Response to Review**
>
> We thank the reviewer for their remarks. We agree that the coupling of graph construction methods with the selection of downstream GNN models is an important topic. To address the reviewer’s recommendation, we have now added an additional section (Section 5.4) which specifically discusses the interplay between graph construction method, graph characteristics/assessment, and downstream model selection. While we agree that an empirical study on the relations between graph construction approaches and resulting graph characteristics is an interesting research area, we believe this to be an original research work and out of scope for a survey paper.
> We agree that applications for brain imaging are highly important research areas and are strongly represented in our survey paper. We believe the reviewed works are proportional to existing lines of work.
>
> Addition to Section 5.4:
> “The graph construction method directly influences the graph structure and therefore the performance of the utilised GNN for the downstream task. We here discuss the interplay between the two components. The graph construction method can be either independent of the graph learning pipeline (static graph construction) or intertwined (adaptive graph construction). In both cases, the choice of graph convolution is an important factor. Zhu et al. (2020) identify some critical design choices that can improve the performance of GNNs on heterophilous graphs. They show that a separate embedding of neighbourhood node features and node-internal features improves the performance of heterophilous graphs as well as a separate embedding of higher-order neighbourhoods and introduces a new model architecture that works well on both homophilous and heterophilous graphs. Graph convolutions that propagate information simultaneously for all neighbours and their own node features are more impacted by heterophilic graphs than convolutions that have separate message-passing schemes for the neighbourhoods and their own node features. There are several graph convolutions, that have been specifically designed for low-homophily graphs, such as H2GCN (Zhu et al., 2020), HEAT convolutions (Mo et al., 2021b), or heterogeneous graph transformers (Hu et al., 2020b). In case the constructed graph results in a low-homophily graph, it is advisable to select one of these graph convolutions for graph learning. However, the choice of such graph convolutions does not guarantee high performance. We furthermore note that graph convolutions that are highly impacted by the graph structure (e.g. GCN (Kipf & Welling, 2016)) benefit most from the utilisation of adaptive graph construction methods. In these cases, the graph structure can be optimised for the whole graph learning pipeline, including the graph convolution at hand. Different models have also been specifically designed for specific tasks. RotatE (Sun et al., 2018), TransE (Bordes et al., 2013), and PairRE (Chao et al., 2021) have, for example, been designed for link prediction tasks and applied for knowledge graph completion. We furthermore identify a systematic evaluation of the connection between model architectures and graph structures as an open research question.”
>
> We furthermore adapted our categorisation scheme for graph types used in medical applications to obtain a clearer distinction as suggested by the reviewer. We added an additional separation between graph types (population-level, subject-level, and generic graph structures) and structure types (concept-based, and spatial). We therefore adapted Figures 2 and 4, added Figure 3 to the manuscript, and discuss the graph types and structure types in the following way:
>
> Addition to Section 3:
> “Medical research and data often contain patient data that defines the structure of the dataset. We identify three distinct graph types that are used in medical applications: (1) population-level graphs, where typically individuals of a cohort are connected in a large graph, (2) subject-level graphs, where each subject is represented by an individual graph –leading to a multi-graph dataset–, and (3) subject-independent graphs, which represent more general structures, such as knowledge graphs, molecules or maps. Each graph type comes with individual challenges and utilises different methods for graph creation. In this section, we give an overview of those three graph types, which are visualised in Figure 2. We furthermore distinguish between two types of structures: (a) relationship-based and (b) spatial structures. Relationship-based structures use concepts and relationships to determine the graph structure and spatial structures use spatial information, for example, image key-points in Euclidean space. All graph types can be combined with all structure types. We summarise the combinations of graph types and structure types with examples in Figure 3.”
>
> [CONTINUED BELOW]

---

> > ### Author Response · Authors · 2023-10-20
> > **Continuation of comment above**
> >
> > [CONTINUED FROM ABOVE]
> >
> > In order to make the recommendations in our work more immediate and accessible, we rephrased some sentences and added additional content to Section 5.
> >
> > Section 5.3:
> > “We here formulate specific recommendations for selecting a graph construction method for different applications and datasets. We discuss the choice of graph type (population-level, subject-level, or subject-independent graphs) as well as the choice between static and adaptive graph construction.”
> >
> > “Selection of graph type The choice between the different graph types is highly dependent on the dataset as well as the desired prediction/task”
> >
> > “Subject-independent graphs are of interest, when no specific relation to a medical subject is available or relevant. This is for example the case when performing link prediction in knowledge
> > graphs, where the relationships between entities (such as diseases and symptoms) are of interest and there is no personal data of patients.“

---

### Review · Reviewer_GrY9 · 2023-10-03

**Summary Of Contributions:**

This paper provides a comprehensive categorization and summary of state-of-the-art graph construction methods in the medical domain. It also offers valuable guidance for selecting the most appropriate graph construction methods based on different datasets.

**Audience:**

Yes

**Claims And Evidence:**

Yes

**Requested Changes:**

Please see weaknesses above

**Strengths And Weaknesses:**

Pros:

1. It provides a comprehensive categorization and summary of state-of-the-art graph construction methods in the medical domain. It also offers valuable guidance for selecting the most appropriate graph construction methods based on different datasets.

2. It also offers valuable guidance for selecting the most appropriate graph construction methods based on different datasets.


Cons:

1. In section 5.3, this paper introduces some metric to assess the constructed graph in terms of similarities in neighborhoods, the author also mentioned that homophily is not a necessary property for successfully graph learning. It is unclear what this paper considers to be a well-constructed graph, higher homophily or lower homophily ratio? If both homophily and heterophily graphs are fine, why do we need to measure the homophily ratio.

2. The recommendations section appears relatively concise in comparison to other sections of the paper. Additionally, some of the recommendations seem to resemble summaries derived from commonly employed methods in existing literature instead of unique insights from the author.

---

> ### Author Response · Authors · 2023-10-20
> **Response to Review**
>
> We thank the reviewer for their insights and recommendations. We extended the discussion on what a “good” graph structure looks like in Section 5.2 in the following way:
>
> “There is no easy answer to what a “good” graph structure looks like. Some graph learning methods are highly sensitive to heterophilic graphs, while others work well on both homophilic and heterophilic graphs (Zhu et al., 2020). We still advise using these graph assessment metrics ideally before (and for adaptive methods also during) model training. They can shed light on the composition of the graph and can potentially reveal reasons for poor performance or guide a suitable graph convolution for performing the downstream task. The evaluation of different metrics that assess different qualities of the graph structure can be essential for understanding the complex interplay between graph structure and model performance.”
>
> Furthermore, we now provide more unique insights regarding recommendations on graph construction methods by extending Section 5. The whole section functions as a set of recommendations for choosing appropriate graph construction methods, both in terms of graph type and static vs. adaptive graph construction methods. We therefore restructured Section 5 in order to make the recommendations regarding the different parts of the graph construction pipeline clearer. In addition, we add Section 5.4, covering some insights into the interplay between graph construction, graph metrics, and graph convolutions.
>
> “The graph construction method directly impacts the graph structure and therefore the performance of the utilised GNN for the downstream task. We here discuss the interplay between the two components. The graph construction method can be either independent of the graph learning pipeline (static graph construction) or intertwined (adaptive graph construction). In both cases, the choice of graph convolution is an important factor. Zhu et al. (2020) identify some critical design choices that can improve the performance of GNNs on heterophilous graphs. They show that a separate embedding of neighbourhood node features and node-internal features improves the performance of heterophilous graphs as well as a separate embedding of higher-order neighbourhoods and introduces a new model architecture that works well on both homophilous and heterophilous graphs. Graph convolutions that propagate information simultaneously for all neighbours and their own node features are more impacted by heterophilic graphs than convolutions that have separate message-passing schemes for the neighbourhoods and their own node features. There are several graph convolutions, that have been specifically designed for low-homophily graphs, such as H2GCN (Zhu et al., 2020), HEAT convolutions (Mo et al., 2021b), or heterogeneous graph transformers (Hu et al., 2020b). In case the constructed graph results in a low-homophily graph, it is advisable to select one of these graph convolutions for graph learning. However, the choice of such graph convolutions does not guarantee high performance. We furthermore note that graph convolutions that are highly impacted by the graph structure (e.g. GCN (Kipf & Welling, 2016)) benefit most from the utilisation of adaptive graph construction methods. In these cases, the graph structure can be optimised for the whole graph learning pipeline, including the graph convolution at hand. Different models have also been specifically designed for specific tasks. RotatE (Sun et al., 2018), TransE (Bordes et al., 2013), and PairRE (Chao et al., 2021) have, for example, been designed for link prediction tasks and applied for knowledge graph completion. We furthermore identify a systematic evaluation of the connection between model architectures and graph structures as an open research question.”

---

### Comment · Reviewer_vdAj · 2023-10-18
**Discussion**

Overall I feel the authors have not done a good job at providing 1. a systematic framework with sufficient levels of details and 2. actionable and deep insights on how to choose between alternatives. For these reasons I’d say they did not achieve what they had proposed and the work should be rejected.

---

### Author Response · Authors · 2023-10-18

Dear TMLR team and reviewers,

there seems to be a misunderstanding, seeing as reviewer vdAJ has now moved to reject our paper while we still not have received a decision on our rebuttal extension request. We kindly asked for an extension for submitting our responses to the reviews until October 20th. We will submit our revised manuscript as discussed in the initial request.

Best,
The authors

---

### Decision · Action_Editor_3raV · 2023-11-22

**Recommendation:** Accept as is

**Comment:**

The paper provides a comprehensive survey of how to apply graph neural networks on medical data, with a main focus on graph construction methods from various kinds of medical data. The reviewers appreciated the thoroughness and breadth of the covered approaches. They raised concerns regarding the use of homophily/heterophily as a metric, and on the insights and recommendations raised by the present survey, but the authors addressed most of their concerned in the revision. In particular, the section on insights and recommendations for graph construction has been significantly extended and improved in the revised version, providing useful signal in the paper beyond just the survey part.

**Audience:**

The paper could be of interest to the ML for medicine community, for a survey of how GNNs may be useful for their data, or too GNN practitioners looking for applications in medicine.

**Claims And Evidence:**

The claims are accurate and clear.